# DUAL-MODE CLOUD-DEVICE COLLABORATION FOR EFFICIENT CONTINUAL ADAPTATION

## ABSTRACT

Continuous environmental changes induce distribution shifts, leading to significant performance degradation of models deployed on resource-constrained mobile devices. Existing fast adaptation methods fail to provide sufficient generalization to meet performance requirements, while cloud-device collaborative learning often relies on a considerable amount of data, limiting real-time applicability. To ensure both timeliness and effectiveness, we propose a dual-mode cloud-device collaborative framework. Specifically, the proposed mothod dynamically switches modes according to the degree of distribution shift: (1) Collaborative adaptation mode handles substantial shifts, where the cloud performs multi-level domain alignment and position-aware prompting to learn domain-invariant representations, which are then distilled to the device model; (2) Self-adaptation mode addresses minor shifts, where the device model performs unsupervised test-time adaptation with pseudo-label generation and quality-aware reweighting for fast local updates. Experimental results show that our framework achieves superior performance while using only 80% of the data and incurring less than 0.5% additional parameters and computation. Moreover, it consistently outperforms compared methods in both accuracy and single-frame inference speed. Code is available at https://anonymous.4open.science/r/DCD-D013.

## 1 INTRODUCTION

Model compression, structured pruning, neural architecture search, compiler-level optimizations, and quantization-aware training have shown great potential in alleviating the inference burden on resource-constrained devices (Cai et al., 2020; Cheng et al., 2024; Zhong et al., 2025). However, these techniques are primarily designed for static data distributions and struggle to maintain stable performance in dynamic environments with continuously shifting distributions or severe data drift. To address this issue, Test-Time Adaptation (TTA) (Liang et al., 2025) has been proposed, which adapts models during inference by leveraging only unlabeled target-domain data, offering a lightweight and efficient solution for real-time deployment. Nevertheless, existing TTA methods still exhibit limited generalization capability when facing large-scale distribution shifts, making them insufficient to fully meet performance requirements.

The existing cloud-device collaboration paradigm relies on powerful cloud models to retrain on new samples and distil the updated knowledge into the lightweight device model. CDCA (Gan et al., 2023) has demonstrated that cloud-based knowledge transfer can effectively enhance the adaptation capability of device models. However, the traditional retraining process entails huge computational costs and involves large-scale data transmission, rendering it challenging to meet real-time requirements. To ensure both timeliness and effectiveness, we introduce a temporal uncertainty–aware dynamic update mechanism that selects different update strategies based on changes in perception tasks. As illustrated in Figure 1, when the estimated uncertainty is below a predefined threshold—indicating minor distribution shifts and low adaptation difficulty—the model is updated locally on the device using limited data. Conversely, when the uncertainty exceeds the threshold—indicating significant distribution shifts and higher adaptation difficulty—the cloud model assists in updating the device model.

Therefore, in the collaborative mode, we build on Domain Faster R-CNN (Chen et al., 2018) then propose Multi-level Domain Adaptation (MuDA). MuDA employs domain discriminators with a

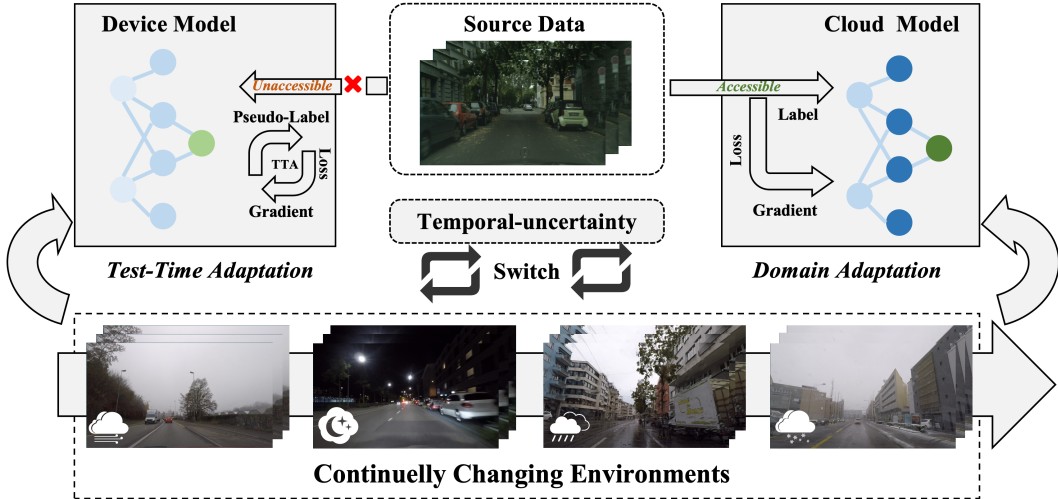

Figure 1: The two modes differ in that the device-side test-time adaptation cannot access source-domain data and can only rely on pseudo-labels generated from test samples. Our dual-mode framework integrates and switches between the two modes based on temporal uncertainty.

gradient reversal layer (Ganin & Lempitsky, 2015) to learn domain-invariant representations, aligning features at the image, instance, and semantic levels. This reduces reliance on large amounts of target-domain data and improves cross-domain generalization. In addition, Position-Aware Prompting (PAP) guides the model to focus on foreground regions and suppress background noise, encouraging attention to domain-invariant areas. Finally, the cloud model's generalized knowledge is transferred to the device model via adapter-based knowledge distillation for efficient updates.

In the self-adaptation mode, the device-side model operates in an offline setting. Since mobile devices lack labeled data but require rapid adaptation to distribution shifts during inference, we propose a quality-aware test-time adaptation (QuTTA) method. QuTTA combines pseudo-labeling with quality-based reweighting to achieve efficient local model updates and incorporates a self-recovery mechanism to mitigate catastrophic forgetting, thereby accelerating adaptation and enhancing stability without the need for additional pseudo-label filtering.

**Main novelty and contributions:** 1) We propose a dual-mode cloud–device continuous adaptation (DMCDA) framework that dynamically switches modes based on temporal uncertainty, enabling mobile models to adapt to evolving environments efficiently. 2) In the self-adaptation mode, we propose QuTTA, which leverages quality-aware reweighting for rapid on-device updates. 3) In the collaborative mode, DMCDA coordinates multiple components to reduce data requirements and communication overhead while preserving strong generalization performance. Experiments demonstrate that DMCDA reduces communication cost by 20% compared with state-of-the-art methods while maintaining superior performance.

## 2 RELATED WORK

**Test-Time Adaptation** aims to enhance model robustness under domain shifts without access to target labels, typically through uncertainty minimization, pseudo-label refinement, and lightweight parameter updates. TENT (Wang et al., 2021) minimizes prediction entropy by updating only BN affine parameters, balancing adaptation and stability. CoTTA (Xu et al., 2022) enhances TENT via EMA ensembling, stochastic restoration, and consistency regularization for long-term adaptation. EATA(Niu et al., 2022) selects reliable, informative samples for entropy minimization updates and applies Fisher regularization to prevent catastrophic forgetting. SAR(Niu et al., 2023) improves stability by removing high-gradient noisy samples and applying sharpness-aware entropy minimization. BeCoTTA Lee et al. (2024) enforces behavior-consistent updates for stable and continual test-time adaptation, while DPCore Zhang et al. (2025) aligns evolving target distributions via dy-

namic prototype correction, and ViDA Liu et al. (2023) leverages low-/high-rank adapters to balance domain-shared and domain-specific knowledge for continual adaptation.

**Cloud-device collaboration** has become an important paradigm for adapting models in dynamic real-world environments. Early works mainly focused on improving computational efficiency (Pacheco et al., 2021; Wang et al., 2022; Guo et al., 2024; Rahmath P et al., 2024), reducing communication overhead (Li et al., 2019; Chen & Ran, 2019; Matsubara et al., 2022; Li et al., 2024; Zhou et al., 2024), or supporting multi-device coordination (Henna & Davy, 2020; Mansour et al., 2021; Ke et al., 2023; Guo et al., 2024). CDCA (Gan et al., 2023) first addressed continual distribution shifts with uncertainty-guided sampling, visual prompts, and EMA-based distillation to enhance on-device robustness. Subsequent works such as AMS (Khani et al., 2021) and ECLM (Zhuang et al., 2023) explored modular model decomposition to enable periodic collaborative updates. Most recently, CDCL (Wang et al., 2024a) extended this concept to multimodal LLMs, where a cloud MLLM guides a lightweight on-device adapter via knowledge distillation and token offloading. These methods show the shift from simple computation offloading to collaborative adaptation and continual learning. CEMA Chen et al. (2024b) further advances this direction by introducing a dual-mode cloud–device continuous adaptation paradigm that dynamically switches between self-adaptation and collaborative adaptation, reducing communication cost while maintaining strong on-device generalization.

# 3 METHODS

## 3.1 OVERVIEW

Traditional uncertainty estimation focuses on a single input by performing Monte Carlo Dropout to compute prediction variance(Gal & Ghahramani, 2016; Guo et al., 2017) , which reflects the model's confidence in handling the current input. While effective for assessing model-level uncertainty, it provides limited insight into whether the environment itself is shifting(Xu et al., 2022; Gan et al., 2023; Nakamura et al., 2024).

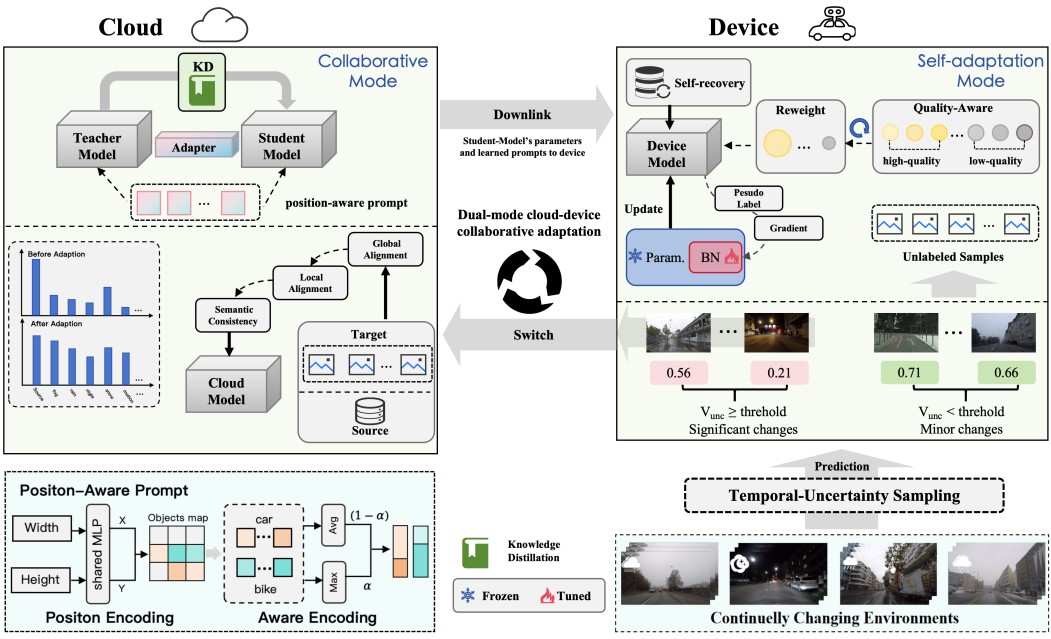

Figure 2: The proposed cloud-device collaborative framework aligns domains using MuDA and PAP, learns domain-invariant features, and transfers knowledge to the device model via distillation. In offline mode, the device model adapts to changing environments through test-time adaptation.

Therefore, in step 1, we estimate the environment-level uncertainty by applying MC Dropout $M$ times on $k$ consecutive frames:

$$V_{\text{unc}} = \sqrt{\frac{1}{kM} \sum_{i=1}^{k} \sum_{m=1}^{M} \|p_m(y \mid x_i) - \bar{p}(y \mid x_i)\|^2}, \qquad (1)$$

where $x_i$ denotes the $i$-th frame, $p_m(y \mid x_i)$ the $m$-th prediction, and $\bar{p}(y \mid x_i)$ the averaged distribution over $M$ predictions. A low $V_{\text{unc}}$ reflects stable conditions and activates local adaptation, while a high value indicates distribution shifts, triggering collaborative mode. We will further explore the impact of different k values and uncertainty thresholds on the framework.

## 3.2 COLLABORATION MODE

Within the cloud collaborative framework, we introduce Multi-level Domain Adaptation (MuDA) and Position-Aware Prompting (PAP) to achieve cross-domain feature alignment.

**MuDA** addresses distribution shifts by performing feature alignment on the teacher model via adversarial training with a Gradient Reversal Layer (GRL) (Ganin & Lempitsky, 2015). Adopting the framework of (Chen et al., 2018), we conduct hierarchical alignment at both the image level ($\mathcal{L}_{\text{img}}$) and instance level ($\mathcal{L}_{\text{ins}}$). Furthermore, to reinforce category-level alignment, we introduce a semantic consistency constraint ($\mathcal{L}_{\text{sc}}$) based on symmetric KL divergence:

$$\mathcal{L}_{\text{sc}} = \sum_{c=1}^{C} \Big[ \text{KL}\big(P_c^s \parallel P_c^t\big) + \text{KL}\big(P_c^t \parallel P_c^s\big) \Big], \qquad (2)$$

where $P_c^s$ and $P_c^t$ denote the class-conditional distributions for category $c$ in source and target domains, respectively. The overall objective is defined as:

$$\mathcal{L}_{\text{total}} = \mathcal{L}_{\text{det}}^{\text{src}} + \lambda_1 \mathcal{L}_{\text{img}} + \lambda_2 \mathcal{L}_{\text{ins}} + \lambda_3 \mathcal{L}_{\text{sc}}, \qquad (3)$$

where $\mathcal{L}_{\text{det}}^{\text{src}}$ is the source supervision loss, and weighting factors $\lambda_{1,2,3}$ are empirically set to 1. Implementation details are provided in Appendix A.3.

**PAP** is proposed to enhance cross-domain adaptability by injecting learnable prompts into the input space under spatial and channel dual guidance. Specifically, we first capture stable structural priors by extracting directional statistics via average pooling and 1D convolution along height and width dimensions, yielding the spatial coordinate weight $W_{\text{coord}}$. Next, to emphasize foreground semantics and suppress background noise, we construct the channel guidance $P_c$ by fusing complementary features from global average ($F_{\text{avg}}$) and max ($F_{\text{max}}$) pooling:

$$P_c = \sigma(\text{MLP}(F_{\text{avg}} + F_{\text{max}})) \cdot F_{\text{avg}} + \big(1 - \sigma(\text{MLP}(F_{\text{avg}} + F_{\text{max}}))\big) \cdot F_{\text{max}}. \qquad (4)$$

Finally, the learnable prompt $\mathbf{P}$ is modulated by these guidance signals and injected into the input image $\mathbf{X}$:

$$\mathbf{X}^* = \mathbf{X} + \mathbf{P} \odot P_c \odot W_{\text{coord}}. \qquad (5)$$

Through this mechanism, PAP explicitly highlights domain-invariant structures at the input level, significantly bolstering the model's cross-domain generalization.

**Knowledge transfer**. After the cloud teacher model learns generalized knowledge from domain adaptation to handle changing environments, it needs to transfer the parameters to the device. Methods like TDD(He et al., 2022) and CrossKD(Wang et al., 2024b) enhance performance by introducing additional mechanisms and modules, but they inevitably increase computational overhead and complexity. To address this issue, we extend the classic CWD(Shu et al., 2021) method by introducing a simple yet effective 1×1 adapter mechanism, which bridges the gap in heterogeneous knowledge distillation while minimizing additional computational costs. Experimental results demonstrate the trade-off between computational load and distillation efficiency, with detailed design and results provided in the Appendix A.5.

## 3.3 LOCAL TTA MODE

In the local self-adaptation mode, we propose QuTTA, a simple yet effective test-time adaptation method. It treats model predictions as pseudo labels for adaptation and employs quality-aware

reweighting to dynamically adjust the strategy based on sample quality. Besides, a self-recovery mechanism is introduced to prevent the model from catastrophic forgetting. Only the parameters of BN layers are updated, ensuring computational efficiency.

**Pseudo-Label Generation.** For an input image $\mathbf{x}$, we directly utilize all detection predictions $\mathcal{D} = \{(\hat{\mathbf{b}}_j, \hat{y}_j, \hat{s}_j)\}_{j=1}^{N}$ as pseudo labels, where $\hat{\mathbf{b}}_j \in \mathbb{R}^4$ represents bounding box coordinates, $\hat{y}_j$ denotes class labels, and $\hat{s}_j$ indicates confidence scores. Unlike conventional approaches, we impose no confidence thresholds or heuristic filtering strategies.

**Quality-Aware Reweighting.** To reduce the impact of noisy pseudo labels, we adopt a quality-aware reweighting strategy that follows the principle: *learn more from high-quality samples (low loss), and less from low-quality ones (high loss)*. We define the reweighting as:

$$\omega(\mathbf{x}) = \max\left(\frac{\tau_{\max}}{1 + \exp(\mathcal{L}_{\det}(\mathbf{x}) - \tau_0)}, \tau_{\min}\right) \tag{6}$$

where $\mathcal{L}_{\det}(\mathbf{x})$ is the detection loss, $\tau_0 = 0.8\ln(1000)$ controls the transition between high and low quality, $\tau_{\max} = 3$ and $\tau_{\min} = 0.4$ set the upper and lower bounds. This function smoothly adjusts weights based on prediction quality, offering stable gradients and avoiding hard threshold tuning. These values may slightly affect the performance of QuTTA; the reasons for their settings and the corresponding ablation studies can be found in the Appendix.

The final test-time adaptation loss incorporates quality-aware weights and temporal regulation:

$$\mathcal{L}_{\text{qutta}}^{(t)}(\mathbf{x}) = \gamma^{(t)} \cdot \omega(\mathbf{x}) \cdot \mathcal{L}_{\det}(\mathbf{x}) \tag{7}$$

where the temporal regulation factor $\gamma^{(t)}$ controls the adaptation strength at iteration $t$. To balance adaptation speed and stability, we adopt a two-stage schedule: the first 40% of samples use a higher intensity ($\gamma = 0.1$) for rapid adaptation, while the remaining 60% use a lower intensity ($\gamma = 0.05$) to ensure stable convergence. We update only the affine parameters of BN layers $\boldsymbol{\phi} = \{\boldsymbol{\gamma}, \boldsymbol{\beta}\}$ while keeping other parameters fixed:

$$\boldsymbol{\phi}^{(t+1)} = \boldsymbol{\phi}^{(t)} - \eta \nabla_{\boldsymbol{\phi}} \mathcal{L}_{\text{qutta}}^{(t)} \tag{8}$$

where $\eta = 5 \times 10^{-7}$ is the learning rate and AdamW (Loshchilov & Hutter, 2017) optimizer is employed for parameter updates.

**Self-Recovery.** To avoid performance degradation caused by accumulated noise during continual adaptation, we introduce a self-recovery mechanism based on monitoring the detection loss with an exponential moving average (EMA).

$$\text{EMA}^{(t)} = \beta \cdot \text{EMA}^{(t-1)} + (1 - \beta) \cdot \mathcal{L}_{\det}^{(t)}, \tag{9}$$

where $\beta$ is the smoothing coefficient. When the EMA value exceeds a pre-defined threshold $\epsilon_{\text{reset}}$ (indicating that the detection loss has significantly increased and performance is deteriorating), the model parameters are reset to the initial state:

$$\text{if} \quad \text{EMA}^{(t)} > \epsilon_{\text{reset}} \quad \Rightarrow \quad \boldsymbol{\theta} \leftarrow \boldsymbol{\theta}_{\text{init}}. \tag{10}$$

Here, $\boldsymbol{\theta}_{\text{init}}$ denotes the initial model parameters, and $\epsilon_{\text{reset}} = 0.3$ means the reset threshold. This design ensures that parameter resetting is only triggered when the model shows clear signs of degradation, rather than when the loss is low, thereby providing a safeguard against catastrophic performance collapse while maintaining stability.

## 4 EXPERIMENTS

### 4.1 DATASETS

We evaluate all methods on severl tasks, using two datasets Cityscapes-C (Cordts et al., 2016) and ACDC-Detection (Sakaridis et al., 2021). To ensure consistency with the benchmark of comparison

Table 1: Benchmark settings for domain adaptation tasks.

| Task | Sequence | Images | Purpose |
|------|----------|--------|---------|
| Task 1 | fog → motion blur → rain → snow → brightness | 2500 × 10 | Evaluation of performance |
| Task 2 | fog → night → rain → snow | 1600 × 10 | Evaluation of performance |
| Task 3 | motion blur → fog → snow | 1800 × 10 | Evaluation of performance and $V_{\text{unc}}$ |
| Task 4 | brightness → rain → night | 1800 × 10 | Evaluation of performance and $V_{\text{unc}}$ |

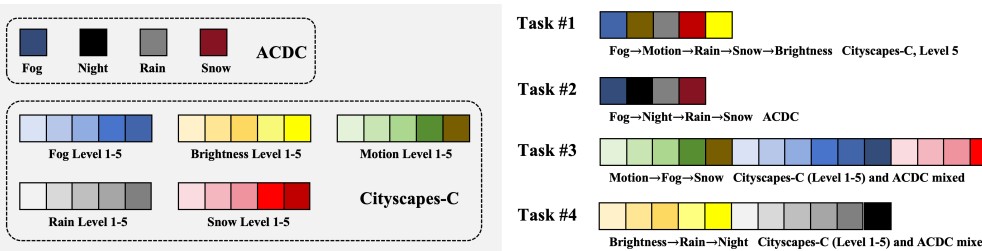

Figure 3: Task sequences composition.

methods and to further evaluate our proposed uncertainty, we design four different tasks, as shown in Table 1 and Figure 2.

The settings of Task 1 and Task 2 remain consistent with CDCA (Gan et al., 2023) and CoTTA (Xu et al., 2022). Task 1 adopts five common dynamic corruptions in real-world scenarios, including brightness, motion blur, rain, fog, and snow, each with five severity levels. Each corrupted subset contains 500 images, and the applied sequence is: fog → motion blur → rain → snow → brightness. Task 2 is derived from the Adverse Conditions Dataset (ACDC), which shares the same categories as Cityscapes, including fog, night, rain, and snow domains. For each adverse condition, 400 unlabeled images are randomly sampled. To further simulate model performance in real environments and evaluate the impact of uncertainty, we design Task 3 and Task 4. For example, Task 3 selects 500 motion samples from Cityscapes-C (with severity levels gradually increasing from 1 to 5, each level contains 100 images), along with fog and snow samples from both datasets (500 from Cityscapes and 100 from ACDC). More details are provided in the appendix.

## 4.2 IMPLEMENTATION DETAILS

In this paper, all experiments are implemented using PyTorch, we follow the default settings in MMdetection(Chen et al., 2019) to train the pre-trained model on the source domain (Cityscapes). On the cloud side, we adopt Faster R-CNN with a ResNet-101 backbone as the large teacher model, while on the device side, we use Faster R-CNN with the ResNet-18 backbone. Other hyperparameters please refer to the appendix.

In Tasks 1 and 2, the primary objective is to evaluate the performance of two modes. To ensure fairness, both TTA and cloud-device collaborative methods use the complete set of test samples, so that the theoretical time required for data upload and download is identical across all compared methods (while ignoring fluctuations caused by real-world network conditions, as this is not the optimization focus of this work). Therefore, the evaluation results solely reflect the model's intrinsic performance and computational efficiency.

## 4.3 RESULTS AND ANALYSIS

We adopt mAP@0.5 (%) as the evaluation metric. For both tasks, we perform 10 rounds of domain adaptation following the aforementioned adaptation order. We report the average performance in the final round and compare it with the source-only baseline, and gain(%) means the improvement of our method compared with Source-only.

**Performance.** Tables 1 and 2 demonstrate that our framework achieves significant advantages in both offline TTA and cloud-device collaborative scenarios. In Task 1, QuTTA improves mAP@50 by 2.7% over the baseline, benefiting from the quality-aware reweighting mechanism that enables rapid

Table 2: Continuous adaptation capability on Cityscapes-to-Cityscapes-C.

| Method | Round 1 | | | | | Round 5 | | | | | Round 10 | | | | | Mean | Gain |
|---|---|---|---|---|---|---|---|---|---|---|---|---|---|---|---|---|---|
| | Fog | Motion | Rain | Snow | Bright | Fog | Motion | Rain | Snow | Bright | Fog | Motion | Rain | Snow | Bright | | |
| *Test-time Adaptation Methods* | | | | | | | | | | | | | | | | | |
| Source | 32.9 | 6.5 | 49.3 | 12.2 | 34.5 | 32.9 | 6.5 | 49.3 | 12.2 | 34.5 | 32.9 | 6.5 | 49.3 | 12.2 | 34.5 | 27.1 | - |
| TENT | 32.9 | 6.5 | **49.5** | 12.2 | 34.7 | 33.8 | 6.9 | 50.2 | 13.1 | 35.9 | 34.4 | 7.6 | **50.5** | 14.4 | 37.0 | 28.8 | +1.7 |
| CoTTA | **33.2** | 5.9 | 48.9 | 12.2 | 34.7 | 33.4 | 6.2 | 49.4 | 12.8 | 35.0 | 33.8 | 6.4 | 50.0 | 13.2 | 35.5 | 27.5 | +0.4 |
| EcoTTA | 32.9 | **6.8** | 49.3 | 12.4 | 34.6 | 33.5 | **7.0** | 50.0 | 12.9 | 35.1 | 33.9 | 7.4 | 50.4 | 13.1 | 35.6 | 28.2 | +1.1 |
| SAR | 33.0 | 6.6 | 49.4 | 12.3 | 34.6 | 33.2 | 6.6 | 49.7 | 12.5 | 34.8 | 33.7 | 6.9 | 50.1 | 12.8 | 34.2 | 27.5 | +0.4 |
| BeCoTTA (S) | 33.4 | 6.7 | 49.6 | 12.5 | 35.1 | 34.2 | 7.1 | 50.4 | 13.4 | 36.2 | 35.0 | 7.5 | 51.0 | 14.4 | 37.3 | 29.0 | +1.9 |
| BeCoTTA (M) | **33.8** | 7.1 | 49.9 | **12.7** | 35.6 | 35.3 | 7.4 | 50.8 | 13.9 | 37.8 | **37.0** | 7.8 | 50.9 | **15.3** | 40.3 | 29.6 | +2.5 |
| BeCoTTA (L) | 33.7 | **7.2** | 50.1 | 12.8 | 35.8 | **35.7** | 7.4 | 50.9 | 14.4 | **38.5** | 37.1 | 7.8 | **51.1** | 15.2 | **41.1** | 30.2 | +3.1 |
| **Ours** | 32.9 | 6.5 | 49.4 | 12.5 | 35.2 | 35.0 | 6.7 | 49.8 | 14.0 | 38.0 | 36.7 | 6.8 | 49.8 | 15.1 | 40.7 | 29.8 | +2.7 |
| *Cloud-device Collaborative Methods* | | | | | | | | | | | | | | | | | |
| AMS | 36.5 | 25.0 | 29.2 | 28.6 | 38.3 | 46.0 | 36.5 | 36.7 | 37.8 | 48.0 | 52.0 | 42.0 | 42.8 | 42.5 | 52.3 | 46.3 | +19.2 |
| CDCA | 39.8 | 27.8 | 31.5 | 30.9 | 41.1 | 50.0 | 40.5 | 40.7 | 41.8 | 52.0 | 52.4 | 47.0 | 47.8 | 47.5 | 57.3 | 50.4 | +23.1 |
| CDCL | 40.5 | 28.2 | 32.0 | 31.4 | 41.6 | 50.8 | 41.2 | 41.5 | 42.6 | 52.7 | 53.8 | 46.8 | 47.5 | 48.3 | 56.5 | 51.6 | +24.5 |
| CoLA | 39.2 | 27.0 | 31.0 | 30.4 | 40.5 | 49.2 | 40.0 | 40.2 | 41.2 | 51.5 | 51.6 | 42.5 | 42.2 | 43.4 | 53.8 | 47.5 | +20.4 |
| CEMA | 39.5 | 27.5 | 31.4 | 30.8 | 40.8 | 49.7 | 40.4 | 40.6 | 41.8 | 51.6 | 52.3 | 44.8 | 45.9 | 46.9 | 55.0 | 48.4 | +21.3 |
| **Ours** | **42.1** | **31.3** | **32.3** | **33.7** | **44.0** | **58.5** | **46.1** | **46.6** | **46.6** | **58.1** | **59.6** | **50.8** | **50.5** | **50.6** | **61.2** | **54.5** | **+27.4** |

Table 3: Continuous generalization capability on Cityscapes-to-ACDC-Detection.

| Method | Round 1 | | | | Round 4 | | | | Round 7 | | | | Round 10 | | | | Mean | Gain |
|---|---|---|---|---|---|---|---|---|---|---|---|---|---|---|---|---|---|---|
| | Fog | Night | Rain | Snow | Fog | Night | Rain | Snow | Fog | Night | Rain | Snow | Fog | Night | Rain | Snow | | |
| *Test-time Adaptation Methods* | | | | | | | | | | | | | | | | | | |
| Source | 35.8 | 10.8 | 16.3 | 22.9 | 35.8 | 10.8 | 16.3 | 22.9 | 35.8 | 10.8 | 16.3 | 22.9 | 35.8 | 10.8 | 16.3 | 22.9 | 21.4 | - |
| TENT | 35.2 | 10.7 | 16.5 | 22.7 | 35.7 | 11.1 | 17.3 | 23.5 | 36.2 | 11.3 | 17.7 | 23.7 | 36.3 | 11.6 | 17.7 | 23.6 | 22.3 | +0.8 |
| CoTTA | 35.7 | 10.9 | 16.2 | 21.9 | 35.1 | 10.8 | 16.1 | 21.8 | 35.1 | 10.9 | 16.1 | 22.0 | 35.6 | 10.9 | 16.2 | 22.3 | 21.2 | -0.2 |
| EcoTTA | **36.0** | 11.1 | 16.8 | 23.1 | 36.4 | 11.4 | 17.2 | 23.6 | 36.9 | 11.6 | 17.6 | 23.8 | 37.0 | 11.7 | 17.8 | 24.0 | 22.5 | +1.1 |
| SAR | 35.4 | 10.5 | 16.1 | 23.2 | 35.6 | 10.6 | 16.1 | 23.3 | 35.7 | 10.7 | 16.2 | 23.7 | 35.8 | 10.8 | 16.3 | 23.4 | 21.6 | +0.1 |
| BeCoTTA (S) | 35.9 | 11.0 | 16.9 | 23.3 | 36.7 | 11.6 | 17.8 | 23.9 | 37.5 | 11.9 | 18.0 | 24.4 | 38.2 | 12.2 | 18.5 | 24.8 | 23.1 | +1.7 |
| BeCoTTA (M) | 36.2 | **11.4** | **17.3** | 23.5 | **37.4** | **12.0** | **18.3** | 24.1 | **38.4** | 12.3 | **18.6** | **25.0** | 38.7 | 12.6 | 18.9 | 25.3 | 23.7 | +2.3 |
| BeCoTTA (L) | 36.1 | 11.3 | 17.2 | 23.6 | 37.2 | 11.9 | 18.3 | **24.7** | 38.2 | **12.5** | 18.7 | 25.0 | **39.2** | **12.9** | **19.1** | **25.5** | **24.2** | **+2.8** |
| **Ours** | 34.9 | 10.5 | 17.0 | 23.0 | 36.4 | 11.5 | 17.9 | 22.4 | 37.6 | 12.1 | 18.1 | 24.8 | 38.8 | 12.8 | 18.9 | 25.1 | 23.9 | +2.5 |
| *Cloud-device Collaborative Methods* | | | | | | | | | | | | | | | | | | |
| AMS | 33.5 | 16.8 | 21.6 | 26.9 | 46.8 | 21.9 | 33.8 | 35.3 | 45.8 | 23.6 | 35.4 | 36.0 | 47.5 | 24.3 | 36.2 | 37.8 | 36.5 | +15.1 |
| CDCA | 34.4 | 17.0 | 21.9 | 27.3 | 46.8 | 21.7 | 33.8 | 35.2 | 48.9 | 23.4 | 36.5 | 36.9 | 50.9 | 25.6 | 39.7 | 40.3 | 39.1 | +17.7 |
| CDCL | 34.4 | 17.2 | 22.2 | 27.6 | 47.5 | 22.1 | 34.5 | 35.9 | 50.1 | 24.1 | 37.6 | 38.1 | 51.8 | 25.7 | 40.5 | 41.6 | 40.2 | +18.8 |
| CoLA | 34.3 | 17.2 | 22.1 | 27.9 | 46.8 | 22.5 | 35.2 | 36.6 | 49.3 | 24.9 | 38.7 | 39.2 | 50.6 | 25.0 | 38.5 | 39.6 | 38.4 | +17.0 |
| CEMA | 34.6 | 17.3 | 22.3 | 28.0 | 47.5 | 22.8 | 35.1 | 37.1 | 50.0 | 24.7 | 38.9 | 39.3 | 51.1 | 25.5 | 39.8 | 40.2 | 39.0 | +18.0 |
| **Ours** | **35.1** | **17.6** | **22.7** | **28.2** | **49.2** | **25.0** | **35.1** | **39.9** | **56.8** | **26.1** | **40.6** | **40.9** | **57.2** | **28.2** | **42.5** | **44.7** | **43.1** | **+21.7** |

response to distribution shifts during inference. Under the cloud-device collaborative mode, our method enhances cross-domain alignment via MuDA and learns domain-invariant features through PAP, thereby enabling efficient knowledge transfer and achieving a remarkable 28.8% improvement over the source-only model, significantly outperforming compared methods. In Task 2, despite larger distribution shifts caused by day-night and weather variations, QuTTA still improves by 2.5% over the baseline and maintains robustness under the challenging night condition; with cloud-device collaboration, it further achieves a 21.7% improvement, outperforming existing approaches.

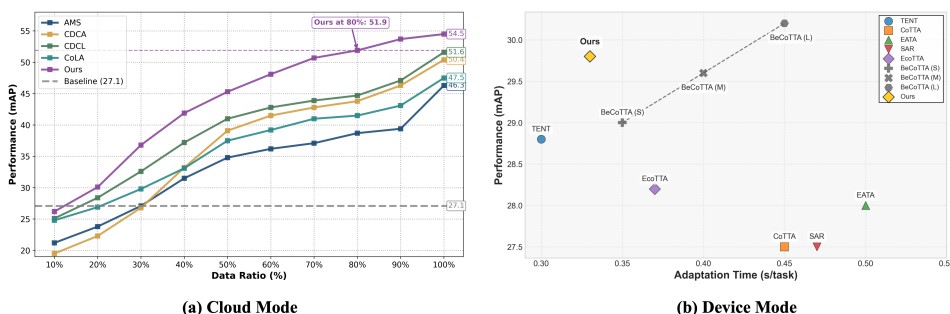

(a) Cloud Mode   (b) Device Mode

Figure 4: (a) Performance under limited target-domain data on Cityscapes-to-Cityscapes-C. (b) Adaptation speed and mAP on Cityscapes-C.

**Efficiency.** In the above tables, all methods use the complete set of test samples. Figure 4 (a) illustrates the performance trends of different methods under varying data ratios. As the amount of target-domain data decreases, the corresponding upload time is reduced; however, most methods exhibit a significant drop in performance. In contrast, our method fully exploits the feature disparities between source and target domains through multi-level domain adaptation, thereby reducing the reliance on large-scale target-domain data while maintaining adaptation efficiency and demonstrating superior data utilization. Notably, even with only 80% of the target-domain data, our method still achieves 51.9 mAP, surpassing all other methods trained with the complete dataset. Therefore, our

framework achieves competitive performance with relatively l ess data, reducing data upload time and, to some extent, improving the overall practicality of the framework.

TTA methods aim to rapidly adapt to distribution shifts during inference, making adaptation speed a key evaluation metric. To comprehensively assess the trade-off between efficiency and accuracy, we adopt task processing speed (s/sample) and mAP for comparative analysis. As shown in Figure 4(b), methods closer to the top-left corner achieve a better balance between efficiency and accuracy. Our QuTTA adapts at 0.36s per sample on Cityscapes-C, ranking second only to TEA in speed while being faster than all other methods, and consistently outperforms them in accuracy, thereby achieving the best overall balance. Additional results on adaptation speed tested on the jetson orin nano platform reported in the Appendix A.7.4.

**Generalization.** To evaluate the sensitivity of our proposed method to different backbone architectures and its ability to generalize across diverse scenes, we test QuTTA on the semantic segmentation setting of the Cityscapes-to-ACDC benchmark **with an input resolution of** $1920 \times 1080$. Aligning with CoTTA and BECoTTA, we use the pretrained SegFormer-B5 as the source model. In addition, two ViT-based approaches, DPCore and ViDA, are included for comparison.

Table 4: Method Sensitivity to Backbone Architecture and Scene Generalization Evaluation on ACDC ($1920 \times 1080$). FPS results are measured or inferred on RTX 4090 and Jetson Orin Nano.

| Method | Round 1 | | | | Round 2 | | | | Round 3 | | | | Mean | Inference time (FPS) | |
| --- | --- | --- | --- | --- | --- | --- | --- | --- | --- | --- | --- | --- | --- | --- | --- |
| | Fog | Night | Rain | Snow | Fog | Night | Rain | Snow | Fog | Night | Rain | Snow | | GPU | Orin |
| Source only | 69.1 | 40.3 | 59.7 | 57.8 | 69.1 | 40.3 | 59.7 | 57.8 | 69.1 | 40.3 | 59.7 | 57.8 | 56.7 | 29.8 | 11.3 |
| TENT | 69.0 | 40.2 | 60.1 | 57.3 | 68.3 | 39.0 | 60.4 | 56.3 | 67.5 | 37.9 | 60.5 | 56.0 | 55.8 | 52.2 | 14.3 |
| CoTTA | 70.9 | 41.2 | 62.4 | 59.7 | 70.9 | 41.1 | 62.0 | 59.4 | 70.7 | 41.0 | 62.7 | 59.5 | 58.5 | 41.3 | 10.2 |
| ViDA | 71.5 | 42.0 | 63.0 | 59.0 | 71.4 | 42.1 | 63.1 | 59.2 | 71.3 | 42.0 | 62.9 | 59.0 | 59.3 | 49.4 | 13.1 |
| DPCore | 72.0 | 42.8 | 63.5 | 60.1 | 72.1 | **42.9** | 63.6 | 60.2 | 72.0 | 42.7 | 63.4 | 60.0 | **61.1** | 51.6 | 13.5 |
| EcoTTA | 68.5 | 35.8 | 62.1 | 57.4 | 68.3 | 35.5 | 62.3 | 57.4 | 68.1 | 35.3 | 62.3 | 57.3 | 55.8 | 46.2 | 11.5 |
| BECoTTA | **71.8** | **48.0** | **66.3** | **62.0** | **71.7** | **47.7** | **66.3** | **61.9** | **71.7** | **47.6** | **66.3** | **61.9** | **61.9** | 48.1 | 12.2 |
| Ours | **71.8** | 46.8 | 65.8 | 61.5 | 71.7 | 46.6 | 65.7 | 61.3 | 71.7 | 46.7 | 65.8 | 61.4 | 61.2 | **54.8** | **15.6** |

From Table 4, we observe that although BECoTTA attains the highest overall mIoU, it incurs a substantial inference-time overhead. In comparison, our method achieves a highly competitive accuracy (mean 61.2), nearly matching BECoTTA, while delivering the fastest inference on both GPU and mobile platforms. This efficiency advantage highlights the superior real-time adaptation capability of our approach, particularly in resource-constrained mobile scenarios.

## 4.4 ABLATION STUDY

In this section, we validate the effectiveness of the proposed MuDA and PAP components through ablation studies and sensitivity analyses, examining their impact on computational and storage efficiency (FLOPs and parameters) as well as the effect of uncertainty on data transmission and performance. Finally, we employ t-SNE visualizations to demonstrate multi-domain alignment and adaptation.

When $(k, M) = (1, 10)$ and the uncertainty threshold is set to 0.3, 38% and 30% of the data in Task 3 and Task 4, respectively, are uploaded to the cloud. The variation in the uploaded data ratio with different $k$ values and uncertainty thresholds is shown in Figure 5 (a).

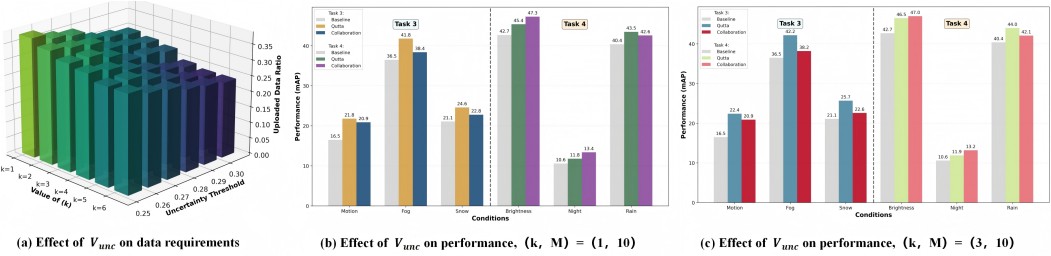

(a) Effect of $V_{unc}$ on data requirements  (b) Effect of $V_{unc}$ on performance, $(k, M) = (1, 10)$  (c) Effect of $V_{unc}$ on performance, $(k, M) = (3, 10)$

Figure 5: Sensitivity analysis of uncertainty $V_{\text{unc}}$, which mainly includes its impact on both data requirements and performance.

Moreover, recalling Eq. 1, when $k = 1$, the uncertainty is estimated by performing $M$ Monte Carlo samples on a single frame; when $k \geq 1$, a sliding window of size $k$ is adopted to incorporate temporal information and measure cross-frame uncertainty. Figures 4(b) and (c) illustrate the performance of the proposed method on Task 3 and Task 4 under the setting of $k = 1, 3$, $M = 10$, and an uncertainty threshold of $0.25$.

In Task 3 and Task 4, QuTTA improves local adaptation performance by approximately 3.4% and 1.5%, respectively, while the collaborative mode remains almost unaffected. This indicates that increasing the window size $k$ mainly benefits local on-device adaptation, whereas the collaborative mode, which relies on cross-domain alignment and cloud knowledge transfer, is less sensitive to the minor data variation introduced by $k$. Additional sensitivity analyses on the uncertainty threshold and window parameters are provided in the Appendix A.8.

We evaluated the impact of the proposed MuDA and PAP components on computational and storage costs as well as detection performance under the collaborative paradigm. The computational overhead is measured by FLOPs, while the storage costs is measured by the parameters (MB). The task is Cityscapes to Cityscapes-C with input image size of $1024 \times 2048$.

Table 5: Ablation study of the proposed components on Cityscapes $\rightarrow$ Cityscapes-C.

| ResNet-18 | ResNet-101 | PAP | MuDA | FLOPs | Params | Mean |
|---|---|---|---|---|---|---|
| ✓ | | | | 300.34 G | 28.32 M | 39.9 |
| ✓ | | ✓ | | 300.64 G | 29.29 M | 41.8 |
| ✓ | | | ✓ | 300.34 G | 28.41 M | 42.0 |
| ✓ | | ✓ | ✓ | 300.64 G | 29.39 M | 43.1 |
| | ✓ | | | 563.75 G | 60.38 M | 52.2 |
| | ✓ | ✓ | | 568.23 G | 66.48 M | 53.6 |
| | ✓ | | ✓ | 563.75 G | 60.48 M | 54.1 |
| | ✓ | ✓ | ✓ | 568.23 G | 66.58 M | 55.8 |

As shown in Table 5, introducing MuDA does not increase FLOPs and only increases the parameters by less than 0.5%, but improves average mAP@50 by 2.1% and 1.9% on ResNet-18 and ResNet-101, respectively. Introducing the PAP component alone also incurs almost no overhead while achieving significant performance improvements. When combining MuDA and PAP, the performance further reaches 3.2% on ResNet-18 and 3.6% on ResNet-101. The results demonstrate that the proposed components can significantly improve model adaptation performance with virtually no increase in resource consumption.

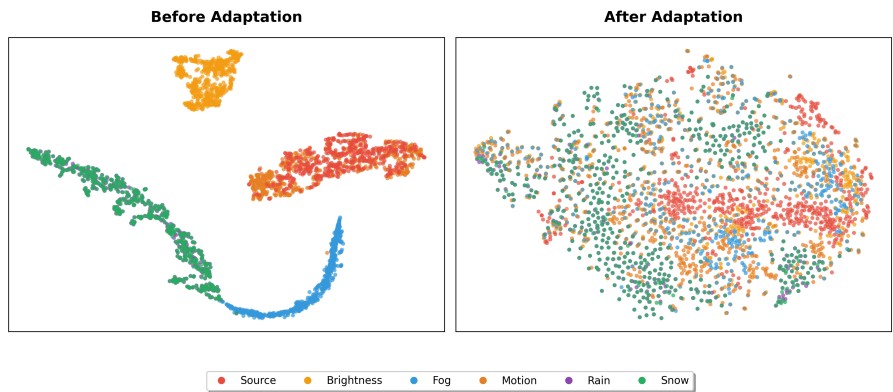

Figure 6: The t-SNE visualization of feature distributions on the Cityscapes-to-Cityscapes-C task before and after applying the domain adaptation strategy.

By comparing the t-SNE visualizations, the effectiveness of MuDA is evident. Before adaptation, a clear domain shift appears, with source and target forming separated clusters. After adaptation, features from both domains overlap substantially, boundaries blur, and distributions align, showing that the strategy effectively learns shared representations and reduces cross-domain discrepancies.

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

# A  APPENDIX

## A.1  LLM USAGE STATEMENT

This paper employed a large language model (LLM) solely for text editing purposes, including grammar correction, fluency improvement, and minor stylistic refinement. The LLM was not used to generate or modify any figures, tables, algorithms, experimental results, or technical content. All scientific contributions, methods, and analyses were developed entirely by the authors.

## A.2 REPRODUCIBILITY STATEMENT

All source code and training logs have been uploaded to an anonymous GitHub repository, which contains no privacy-sensitive information or identifiable links.

## A.3 MORE DETAILS OF MuDA

### A.3.1 GRL MECHANISM

In the MuDA, we introduce the GRL to facilitate domain adaptation through adversarial training. The GRL helps in learning domain-invariant features by reversing the gradient during backpropagation. Specifically, during the forward pass, GRL acts as an identity function, meaning it does not modify the input. However, during the backpropagation, it reverses the gradient by multiplying it by a negative scalar factor $-\lambda$. This reversal of the gradient forces the feature extractor to learn representations that are domain-invariant.

Mathematically, the GRL operates as follows:

- **Forward Pass:**

$$R_\lambda(\mathbf{x}) = \mathbf{x}$$

- **Backward Pass:**

$$\frac{dR_\lambda(\mathbf{x})}{d\mathbf{x}} = -\lambda I$$

Where $R_\lambda(\mathbf{x})$ is the operation performed by GRL, $\mathbf{x}$ is the input feature, $I$ is the identity matrix, and $\lambda$ is the scalar that controls the strength of the gradient reversal. By using the GRL, the feature extractor is encouraged to learn features that are not only discriminative for the source domain but also invariant to domain shifts, thereby enabling better adaptation to the target domain.

### A.3.2 LOSS FUNCTION OF MuDA

The total loss function used in the MuDA framework is a weighted sum of several loss components, each designed to achieve a specific goal. The loss components include the label prediction loss on the source domain, image-level alignment loss, instance-level alignment loss, and semantic consistency loss. The total loss function is defined as:

$$\mathcal{L}_{\text{total}} = \mathcal{L}_{\text{det}}^{\text{src}} + \lambda_1 \mathcal{L}_{\text{img}} + \lambda_2 \mathcal{L}_{\text{ins}} + \lambda_3 \mathcal{L}_{\text{sc}},$$

where, $\mathcal{L}_{\text{det}}^{\text{src}}$ is the supervised detection loss on the source domain, $\mathcal{L}_{\text{img}}$ is the image-level alignment loss that mitigates global differences such as style and illumination between domains, $\mathcal{L}_{\text{ins}}$ is the instance-level alignment loss that focuses on aligning local object features, $\mathcal{L}_{\text{sc}}$ is the semantic consistency loss that enforces cross-domain semantic alignment.

The source domain label prediction loss $\mathcal{L}_{\text{det}}^{\text{src}}$ is a standard supervised loss, typically the cross-entropy loss, which measures the difference between the predicted class probabilities and the true class labels. It ensures that the model learns to correctly classify the source domain samples. The loss is defined as:

$$\mathcal{L}_{\text{det}}^{\text{src}} = -\sum_{i=1}^{N} y_i \log(\hat{y}_i),$$

where, $y_i$ is the true label of the $i$-th sample from the source domain, $\hat{y}_i$ is the predicted label for the $i$-th sample, $N$ is the number of source domain samples.

The image-level alignment loss $\mathcal{L}_{\text{img}}$ is designed to reduce the global differences between the source and target domains, such as illumination and style. This is done by aligning the feature representations of the entire images from both domains. The loss is defined as:

$$\mathcal{L}_{\text{img}} = \frac{1}{N} \sum_{i=1}^{N} \|\mathbf{f}(\mathbf{x}_i^s) - \mathbf{f}(\mathbf{x}_i^t)\|_2^2,$$

where, $\mathbf{x}_i^s$ and $\mathbf{x}_i^t$ represent the $i$-th image from the source and target domains, respectively, $\mathbf{f}(\cdot)$ is the feature extractor, $\|\cdot\|_2$ is the Euclidean distance, which measures the similarity between feature representations.

The instance-level alignment loss $\mathcal{L}_{\text{ins}}$ focuses on aligning the features of local object instances in the source and target domains. This loss helps to ensure that the model learns similar representations for corresponding objects across domains. The formula is:

$$\mathcal{L}_{\text{ins}} = \sum_{i=1}^{N} \|\mathbf{f}(\hat{\mathbf{b}}_i^s) - \mathbf{f}(\hat{\mathbf{b}}_i^t)\|_2^2,$$

where, $\hat{\mathbf{b}}_i^s$ and $\hat{\mathbf{b}}_i^t$ represent the $i$-th object bounding boxes in the source and target domains, $\mathbf{f}(\hat{\mathbf{b}}_i)$ is the feature extractor applied to the bounding boxes.

### A.4 OPTIMIZATION AND BACKPROPAGATION

The MuDA framework is optimized using standard backpropagation. The parameters of the feature extractor, label predictor, and domain classifier are updated using stochastic gradient descent (SGD). During backpropagation, the gradient reversal layer multiplies the gradient by $-\lambda$, ensuring that the feature extractor learns domain-invariant features while the domain classifier tries to distinguish between source and target domains.

The gradients for the feature extractor and the domain classifier are updated as follows:

- Feature Extractor Update:

$$\theta_f \leftarrow \theta_f - \mu \left( \frac{\partial \mathcal{L}_y}{\partial \theta_f} - \lambda \frac{\partial \mathcal{L}_d}{\partial \theta_f} \right)$$

- Label Predictor Update:

$$\theta_y \leftarrow \theta_y - \mu \frac{\partial \mathcal{L}_y}{\partial \theta_y}$$

- Domain Classifier Update:

$$\theta_d \leftarrow \theta_d - \mu \frac{\partial \mathcal{L}_d}{\partial \theta_d}$$

Where $\mu$ is the learning rate, and $\mathcal{L}_y$ and $\mathcal{L}_d$ represent the label prediction loss and domain classification loss, respectively.

### A.5 ADPTER-BASED DISTILLATION

Channel-Wise Distillation (CWD) aligns the feature distributions of teacher and student models at the channel level, enabling more fine-grained knowledge transfer compared to global feature or output alignment. However, in heterogeneous distillation, the significant architectural differences between teacher and student models often lead to unstable training or even gradient explosion when directly enforcing feature alignment. To address this issue, a lightweight adapter (e.g., a $1 \times 1$ convolution) is commonly introduced to bridge the feature discrepancy, which not only ensures dimensional compatibility and stabilizes gradients but also filters transferable knowledge. As a result, the adapter substantially enhances the stability and effectiveness of heterogeneous distillation.

Specifically, we design a lightweight adapter using $1 \times 1$ convolution followed by BN and ReLU to map student features into the teacher space:

$$L_{\text{adapter}} = \frac{1}{BCHW} \left\| F^{(t)} - A(F^{(s)}) \right\|_2^2, \tag{11}$$

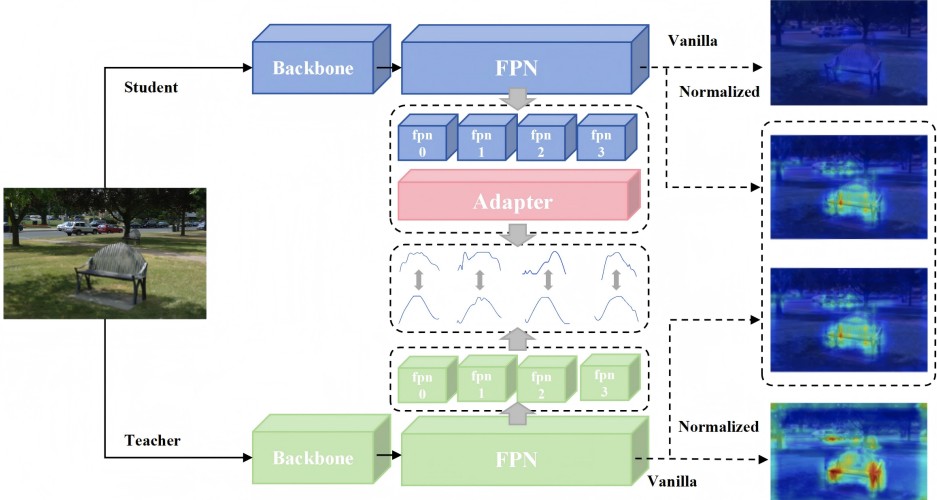

Figure 7: The design of adpter-based cwd.

where $F^{(t)}$ and $F^{(s)}$ denote the teacher and student feature maps of dimension $B \times C \times H \times W$, and $A(\cdot)$ is a lightweight adapter implemented by a $1 \times 1$ convolution followed by BN and ReLU, which maps student features into the teacher space.

$$L_{\text{cwd}} = \frac{1}{BHW} \sum_{c=1}^{C} P_c^{(t)} \log \frac{P_c^{(t)}}{P_c^{(s)}}, \tag{12}$$

where $P_c^{(t)}$ and $P_c^{(s)}$ are normalized activation distributions of the $c$-th channel for teacher and student, respectively, encouraging alignment of channel-wise activation patterns.

$$L_{\text{d}} = \lambda_{\text{adapter}} L_{\text{adapter}} + \lambda_{\text{cwd}} L_{\text{cwd}}, \tag{13}$$

where $\lambda_{\text{adapter}}$ and $\lambda_{\text{cwd}}$ are trade-off coefficients that balance the contributions of adapter alignment and channel-wise distillation to the final objective.

## A.6 DETAILS OF TASKS

For dataset configurations, we strictly follow the setups of CDCA and CoTTA. For local test-time adaptation, the AdamW optimizer is used with a learning rate of 5e-7, while the cloud model is optimized using SGD.

**Cityscapes → Cityscapes-C.** The Cityscapes-C dataset was originally designed for robustness evaluation (Hendrycks & Dietterich, 2019). We select five common types of dynamic scene corruptions frequently encountered in real-world environments, including brightness, motion blur, rain, fog, and snow. Each corruption has five severity levels, and the corruptions are applied to the validation images of the Cityscapes dataset. Each corrupted dataset contains 500 images, and the applied sequence is: fog → motion blur → rain → snow → brightness. For this task, we follow the configuration in MMDetection (Chen et al., 2019) to train a pre-trained source model on the Cityscapes dataset.

**Cityscapes → ACDC-Detection.** The ACDC-Detection dataset is derived from the Adverse Conditions Dataset (ACDC), which shares the same classes as Cityscapes, including fog, night, rain, and snow domains. We utilize the official detection annotations and follow the task setting in CoTTA. During adaptation, we randomly sample 400 unlabeled images for each adverse condition. To simulate scenarios where the model revisits similar environments and to evaluate the continual generalization ability of our approach, we repeat the sequence of four conditions 10 times (in the order: fog

→ night → rain → snow → fog ...). For this task, the source model is trained in the same manner as in the Cityscapes → Cityscapes-C benchmark.

**Task 3.** Task 3 is designed to simulate more natural transitions between adverse conditions by mixing motion blur, fog, and snow. Specifically, we select 500 motion blur samples from Cityscapes-C, where the corruption severity levels gradually increase from 1 to 5 with 100 images for each level. In addition, we include 500 fog samples from Cityscapes and 100 fog samples from ACDC, together with 500 snow samples from Cityscapes and 200 snow samples from ACDC, resulting in 1800 images in total. The corruption sequence is applied in the order: motion blur → fog → snow. This ordering mimics realistic scenarios where dynamic blur is often followed by low-visibility conditions such as fog, eventually leading to snow. The task is repeated 10 times to evaluate continual adaptation performance and the sensitivity of the uncertainty measure $V_{\mathrm{unc}}$.

**Task 4.** Task 4 focuses on brightness, rain, and night conditions to emulate illumination changes that occur naturally over time. We select 500 brightness-corrupted images from Cityscapes-C (again with severity levels from 1 to 5, each level containing 100 images), 500 rain samples from Cityscapes, 300 rain samples from ACDC, and 500 night samples from ACDC, resulting in 1800 images in total. The sequence is arranged as brightness → rain → night, which simulates the gradual transition from daytime with varying brightness, to rainy weather, and finally to nighttime conditions. This sequence not only reflects realistic environmental evolution but also stresses the robustness of models when facing compounded illumination and weather shifts. The task is repeated 10 times for continual evaluation of both performance and $V_{\mathrm{unc}}$.

### A.7 MORE EXPERIMENTAL DETAILS

#### A.7.1 BASELINE

We compare our approach with a range of baselines covering TTA and cloud-device collaboration methods. Source-only directly evaluates the source-pretrained model on the target domain without adaptation. TTA methods include TENT (Wang et al., 2021), CoTTA (Xu et al., 2022), EATA (Niu et al., 2022), SAR (Niu et al., 2023), BeCoTTA Lee et al. (2024) and ReCAP (Hu et al., 2025), while cloud-device collaboration methods include Pseudo-label, AMS (Khani et al., 2021), CDCL (Gan et al., 2023), CDCA (Wang et al., 2024a) and CoLA (Chen et al., 2024a). builds upon CDCA by introducing adapters and knowledge distillation to transfer generalization knowledge between large and small models

#### A.7.2 MORE RESULTS AND ANALYSIS OF TASK 2

In Section 4.3, we analyzed the efficiency of different TTA methods on Task 1 (Cityscapes → Cityscapes-C). The results showed that as the proportion of target-domain data decreases, most methods experience noticeable accuracy drops despite reduced upload time. In contrast, QuTTA maintains stable adaptation efficiency and achieves superior performance even with limited data.

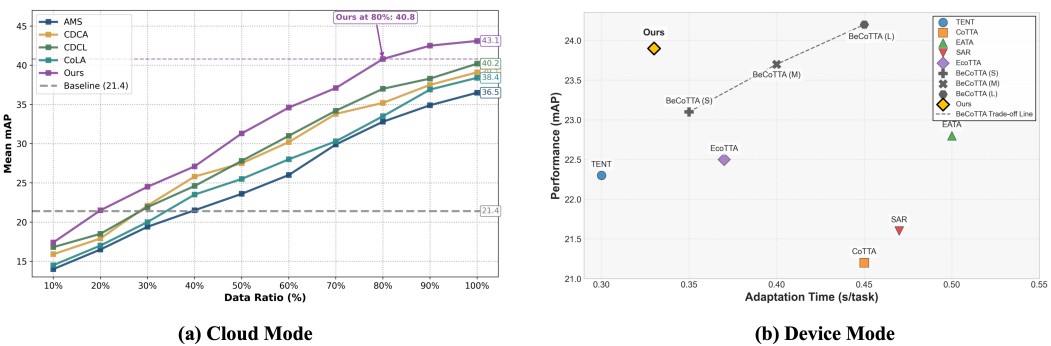

(a) Cloud Mode          (b) Device Mode

Figure 8: Performance under limited target-domain data on ACDC.

Following the same procedure, we extend the evaluation to Task 2 (Cityscapes → ACDC-Detection). As shown in Figure 8, QuTTA consistently achieves competitive performance while keeping adap-

Table 6: Continuous adaptation capability on Task 3 (Motion→Fog→Snow).

| Method | Round 1 | | | Round 5 | | | Round 10 | | | Mean | Gain |
|--------|---------|-----|------|---------|-----|------|----------|-----|------|------|------|
| | Motion | Fog | Snow | Motion | Fog | Snow | Motion | Fog | Snow | | |
| *Test-time Adaptation Methods* | | | | | | | | | | | |
| Source | 19.5 | 38.6 | 21.6 | 19.5 | 38.6 | 21.6 | 19.5 | 38.6 | 21.6 | 26.6 | - |
| TENT | 19.4 | 38.5 | 21.4 | 19.8 | 38.9 | 22.1 | 20.3 | 39.2 | 22.8 | 26.9 | +0.3 |
| CoTTA | 19.2 | **38.8** | 21.5 | 19.6 | 38.7 | 21.9 | 19.9 | 38.9 | 22.4 | 26.8 | +0.2 |
| EATA | 19.5 | 38.6 | 21.6 | **20.1** | 38.9 | 22.2 | 20.5 | 39.1 | 22.6 | 27.0 | +0.4 |
| SAR | **19.6** | 38.7 | 21.7 | 19.8 | 38.8 | 22.0 | 20.1 | 38.9 | 22.3 | 26.9 | +0.3 |
| TEA | 19.3 | 38.4 | 21.4 | 19.1 | 37.8 | 21.1 | 18.8 | 37.2 | 20.6 | 25.9 | -0.7 |
| ReCAP | 19.4 | 38.6 | 21.8 | 20.0 | **39.3** | **22.5** | 20.6 | **39.6** | 23.1 | 27.2 | +0.6 |
| **QuTTA (Ours)** | 19.6 | 38.7 | **21.9** | 21.2 | 40.4 | 23.4 | 23.6 | 42.7 | 25.3 | **28.5** | **+1.9** |
| *Cloud-device Collaborative Methods* | | | | | | | | | | | |
| AMS | 26.8 | 35.2 | 28.4 | 34.1 | 44.8 | 36.2 | 38.5 | 48.3 | 40.1 | 36.9 | +10.3 |
| CDCA | 28.1 | 36.9 | 30.2 | 36.4 | 47.2 | 38.5 | 39.8 | 50.1 | 41.8 | 38.8 | +12.2 |
| CDCL | 28.5 | 37.3 | 30.8 | 37.1 | 47.8 | 39.1 | 40.2 | 50.6 | 42.3 | 39.3 | +12.7 |
| CoLA | 27.9 | 36.7 | 30.1 | 35.8 | 46.9 | 38.2 | 39.1 | 49.7 | 41.5 | 38.4 | +11.8 |
| **Ours** | **32.4** | **40.2** | **34.5** | **38.9** | **48.7** | **40.8** | **40.2** | **52.7** | **42.5** | **41.2** | **+14.6** |

tation latency low. These results confirm that our method remains efficient across different target domains, maintaining robust performance under varying levels of target-domain data availability.

### A.7.3 MORE RESULTS AND ANALYSIS OF TASK 3 AND 4

Different from the ablation study in section 4.3 where only a portion of target data was uploaded, in this section we follow the main paper setting and load 100% of the target-domain data for all methods. In this way, the impact of data uploading/downloading latency is eliminated, and the evaluation focuses solely on the model performance. Tables 6 and 7 further validate the effectiveness of our framework under more natural corruption sequences.

In Task 3 (Motion → Fog → Snow), QuTTA improves mAP@50 by 1.9% over the source-only baseline, demonstrating its ability to adaptively reweight pseudo-labels and stabilize training even when facing dynamic degradations such as motion blur followed by visibility changes. Under the cloud-device collaborative setting, our method achieves a remarkable 14.6% improvement, significantly outperforming existing approaches by leveraging cross-domain alignment and knowledge transfer.

Table 7: Continuous generalization capability on Task 4 (Brightness→Rain→Night).

| Method | Round 1 | | | Round 5 | | | Round 10 | | | Mean | Gain |
|--------|---------|-----|-------|---------|-----|-------|----------|-----|-------|------|------|
| | Bright | Rain | Night | Bright | Rain | Night | Bright | Rain | Night | | |
| *Test-time Adaptation Methods* | | | | | | | | | | | |
| Source | 42.5 | 13.4 | 43.6 | 42.5 | 13.4 | 43.6 | 42.5 | 13.4 | 43.6 | 33.2 | - |
| TENT | 42.3 | 13.6 | 43.4 | 42.8 | 14.1 | 43.9 | 43.2 | 14.5 | 44.3 | 33.6 | +0.4 |
| CoTTA | 42.1 | 13.2 | 43.5 | 42.4 | 13.5 | 43.7 | 42.7 | 13.8 | 44.1 | 33.3 | +0.1 |
| EATA | 42.5 | **13.8** | **43.8** | 42.9 | 14.2 | 44.1 | 43.1 | 14.6 | 44.5 | 33.7 | +0.5 |
| SAR | 42.3 | 13.3 | 43.4 | 42.6 | 13.6 | 43.7 | 42.8 | 13.9 | 44.0 | 33.3 | +0.1 |
| TEA | 42.2 | 13.5 | 43.7 | 41.8 | 13.2 | 43.4 | 41.5 | 12.9 | 43.1 | 32.8 | -0.4 |
| ReCAP | 42.5 | **13.8** | **43.8** | 42.9 | 14.2 | 44.1 | 43.1 | 14.6 | 44.5 | 33.7 | +0.5 |
| **QuTTA (Ours)** | **42.8** | 13.5 | 43.9 | 45.6 | **15.2** | 44.8 | 49.4 | 17.1 | 46.2 | **35.4** | **+2.2** |
| *Cloud-device Collaborative Methods* | | | | | | | | | | | |
| AMS | 38.3 | 18.5 | 39.2 | 47.1 | 26.8 | 48.5 | 52.8 | 30.1 | 53.4 | 39.4 | +6.2 |
| CDCA | 39.8 | 19.2 | 40.6 | 49.5 | 28.3 | 50.8 | 55.7 | 31.8 | 56.2 | 41.3 | +8.1 |
| CDCL | 40.1 | 19.5 | 41.1 | 50.2 | 28.9 | 51.5 | 56.3 | 32.4 | 56.9 | 41.8 | +8.6 |
| CoLA | 39.5 | 19.1 | 40.3 | 48.8 | 28.1 | 50.2 | 55.1 | 31.5 | 55.8 | 40.9 | +7.7 |
| **Ours** | **48.2** | **22.4** | **48.8** | **56.3** | **29.1** | **54.2** | **60.3** | **33.8** | **57.5** | **45.6** | **+12.4** |

In Task 4 (Brightness → Rain → Night), where distribution shifts are dominated by illumination changes and severe night scenes, QuTTA still yields a 2.2% improvement over the baseline, confirming its robustness to compounded weather and lighting variations. With cloud-device collaboration, it further enhances adaptation by aligning brightness and rain-induced distortions with the night domain, resulting in a substantial 12.4% improvement over the source model, again surpassing compared methods.

Table 8: Comparison of adaptation time (s/task) and performance (mAP) between RTX A6000 and Jetson Orin Nano 8GB.

| Dataset | Method | mAP | A6000 | Orin |
|---|---|---|---|---|
| ACDC | TENT | 21.2 | 0.26 | 0.99 |
| | CoTTA | 20.1 | 0.29 | 1.03 |
| | EATA | 21.1 | 0.28 | 1.01 |
| | SAR | 20.4 | 0.27 | 0.94 |
| | **TEA** | 20.9 | **0.21** | **0.82** |
| | ReCAP | 21.5 | 0.26 | 1.05 |
| | **QuTTA** | **22.7** | 0.24 | 0.94 |
| Cityscapes | TENT | 27.6 | 0.40 | 1.50 |
| | CoTTA | 25.9 | 0.67 | 2.33 |
| | EATA | 27.0 | 0.50 | 1.85 |
| | SAR | 26.3 | 0.45 | 1.58 |
| | **TEA** | 25.1 | **0.31** | **1.24** |
| | ReCAP | 28.2 | 0.38 | 1.52 |
| | **QuTTA** | **28.7** | 0.36 | 1.21 |

### A.7.4 EFFICIENCY

When deploying test-time adaptation (TTA) across heterogeneous hardware platforms, the computational disparity between devices becomes a critical factor that directly impacts adaptation efficiency and practical applicability. In section4.3, we test methos on RTX A6000, which deliver over 38 TFLOPS of FP32 throughput and hundreds of GB/s memory bandwidth, enabling sub-second adaptation per task. In contrast, low-power edge devices such as the Jetson Orin Nano 8GB provide only around 40 INT8 TOPS and 68 GB/s bandwidth, resulting in nearly an order of magnitude lower effective compute. Evaluating TTA methods under both desktop/server-class and embedded platforms is therefore essential to understand their feasibility in real-world scenarios such as autonomous driving, robotics, and edge perception.

Table 8 shows the adaptation time (s/task) and performance (mAP) on the ACDC and Cityscapes benchmarks. On the RTX A6000, all methods achieve adaptation within 0.2–0.6 seconds per task, while on the Orin Nano the latency expands to 1–2.5 seconds per task, confirming the significant overhead incurred on resource-constrained devices. Furthermore, the relative differences between methods become more pronounced in the low-compute setting: for instance, on Cityscapes, CoTTA requires 2.33 seconds per task whereas TEA only takes 1.24 seconds. In terms of accuracy, QuTTA consistently achieves the highest mAP (22.7 and 28.7) across both benchmarks, while TEA demonstrates the lowest latency, highlighting the trade-off between efficiency and robustness under heterogeneous deployment conditions.

### A.8 MORE ABLATION STUDY

The choice of $k$ and the uncertainty $V_{\text{unc}}$ influences the volume of data uploaded to the cloud, thereby affecting performance. With the number of MC samples fixed at 10, we vary $k$ and the threshold to investigate their impact through sensitivity analysis.

In Tasks 3 and 4, increasing the window size $k$ leads to gradual performance gains for QuTTA, which eventually stabilizes. For example, in Task 3, Motion improves from 21.8 ($k = 1$) to 23.1 ($k = 10$), Fog from 41.8 to 43.3, and Snow from 24.6 to 27.3. Similarly, in Task 4, Brightness rises from 45.4 to 47.7, Night from 11.8 to 12.3, and Rain from 43.5 to 45.2. These results indicate that cross-frame information enhances on-device adaptation. In contrast, the collaborative mode remains largely unaffected or slightly degrades, e.g., Task 3 Motion decreases from 20.9 to 20.7 and Fog from 38.4 to 38.1, while in Task 4 Brightness drops from 47.3 to 46.5 and Rain from 42.6 to 41.1, reflecting its reliance on cross-domain alignment and knowledge transfer rather than local temporal parameters.

Figure 9 shows the effect of varying the uncertainty threshold from 0.25 to 0.30 on Task 3. For QuTTA, performance degrades steadily as the threshold increases (e.g., Motion $21.8 \rightarrow 20.6$, Fog $41.8 \rightarrow 40.1$, Snow $24.6 \rightarrow 23.5$), indicating that local adaptation is highly sensitive to threshold

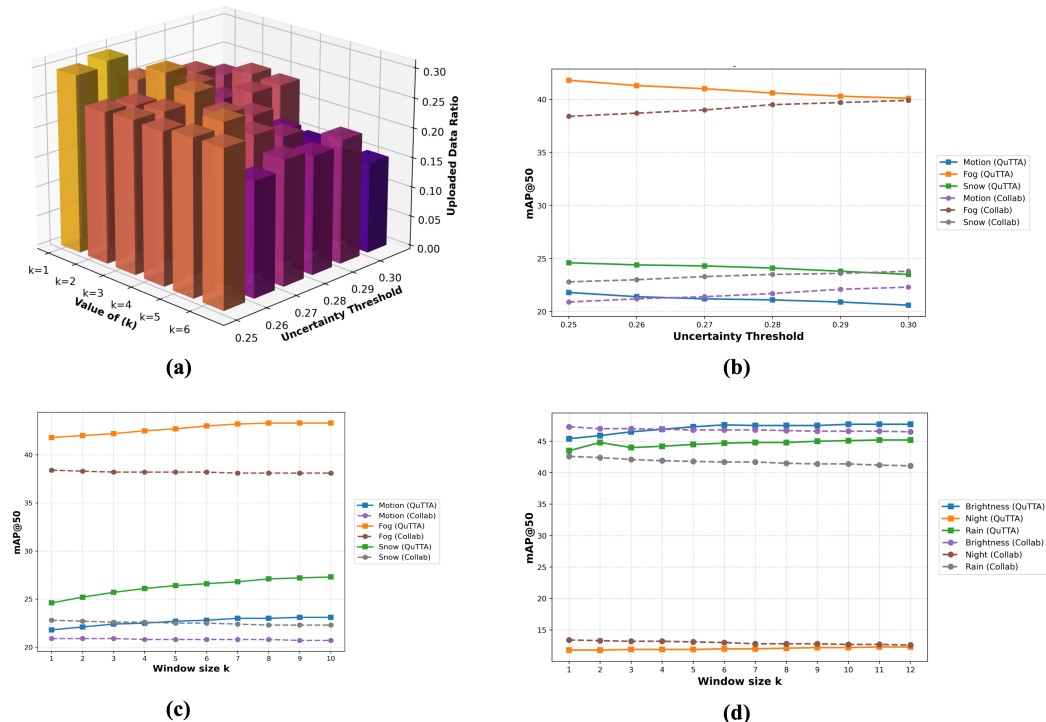

(a)                        (b)

(c)                        (d)

Figure 9: (a) Sensitivity analysis of data requirements with respect to uncertainty settings (b) Impact of uncertainty threshold on performance (Task 3). (c) Impact of k value on performance (Task 3). (d) Impact of k value on performance (Task 4).

choice. In contrast, the collaborative mode is much less affected, with only minor variations across tasks (e.g., Motion $20.9 \to 22.3$, Fog $38.4 \to 39.9$, Snow $22.8 \to 23.8$). Based on this observation, we adopt $\tau = 0.25$, which yields the best overall performance while avoiding unnecessary degradation in local adaptation.

### A.9   SWITCHING AND STABILITY ANALYSIS

To evaluate the stability of the dynamic switching mechanism, Table 9 reports the data allocation and corresponding performance across 10 rounds of continuous adaptation. The total number of images is fixed at 1800 per round, while both device-side and cloud-side mAP values exhibit mild fluctuations rather than monotonic increases, reflecting realistic stochastic behavior. All variations remain within tight bounds: the device-side mAP fluctuates within 0.72%, and the cloud-side within 0.98%, indicating strong robustness of the switching mechanism.

The allocation ratios remain highly stable, fluctuating only within $\pm 1.5\%$. Despite the stochasticity introduced by MC Dropout, the system preserves consistent behavior across rounds. The lightly oscillating mAP curves further demonstrate that occasional switching variances do not deteriorate performance, confirming the reliability and robustness of the dual-mode adaptation framework.

### A.10   MORE ANALYSIS ABOUT THE EFFECT OF RESOLUTION

In addition to the standard resolution evaluations presented in the main text, we provide a supplementary analysis on the Cityscapes-to-ACDC benchmark using a **reduced input resolution of** $960 \times 540$. This experiment aims to demonstrate the system's capability in resource-constrained scenarios where inference speed is prioritized. Consistent with the main experiments, we employ the pretrained SegFormer-B5 as the source backbone and benchmark against state-of-the-art methods, including CoTTA, BECoTTA, and ViT-based approaches (DPCore and ViDA).

Table 9: Sample switching and detection performance over 10 rounds.

| Round | Device (Kept) | Cloud (Uploaded) | Device mAP | Cloud mAP |
|---|---|---|---|---|
| R1 | 1180 | 620 | 27.40% | 39.10% |
| R2 | 1165 | 635 | 27.15% | 39.55% |
| R3 | 1158 | 642 | 27.50% | 38.90% |
| R4 | 1148 | 652 | 27.68% | 39.35% |
| R5 | 1139 | 661 | 27.32% | 38.80% |
| R6 | 1152 | 648 | 27.82% | 39.25% |
| R7 | 1160 | 640 | 27.60% | 39.75% |
| R8 | 1170 | 630 | 27.48% | 39.05% |
| R9 | 1155 | 645 | 27.85% | 39.50% |
| R10 | 1143 | 657 | 27.68% | 39.08% |
| **Range** | – | – | **0.72%** | **0.98%** |
| **Avg. Fluctuation** | $\pm1.5\%$ | $\pm1.5\%$ | $\pm0.28\%$ | $\pm0.41\%$ |

Table 10: Supplementary Evaluation on ACDC with Reduced Resolution ($960 \times 540$). This setting simulates edge deployment scenarios. FPS results are measured on RTX 4090 and Jetson Orin Nano.

| Method | Round 1 | | | | Round 2 | | | | Round 3 | | | | Mean | Inference time (FPS) | |
|---|---|---|---|---|---|---|---|---|---|---|---|---|---|---|---|
| | Fog | Night | Rain | Snow | Fog | Night | Rain | Snow | Fog | Night | Rain | Snow | | GPU | Orin |
| Source only | 66.5 | 37.8 | 56.4 | 54.2 | 66.5 | 37.8 | 56.4 | 54.2 | 66.5 | 37.8 | 56.4 | 54.2 | 53.7 | 65.2 | 24.5 |
| TENT | 66.2 | 37.5 | 57.1 | 53.8 | 65.8 | 36.4 | 57.5 | 53.0 | 65.1 | 35.2 | 57.6 | 52.8 | 53.2 | 105.4 | 31.2 |
| CoTTA | 68.4 | 38.6 | 59.5 | 56.4 | 68.3 | 38.5 | 59.2 | 56.1 | 68.1 | 38.4 | 59.8 | 56.2 | 55.6 | 88.6 | 22.4 |
| ViDA | 68.9 | 39.5 | 60.1 | 56.0 | 68.8 | 39.6 | 60.2 | 56.2 | 68.7 | 39.5 | 60.0 | 56.0 | 56.5 | 98.5 | 28.6 |
| DPCore | 69.4 | 40.1 | 60.6 | 57.2 | 69.5 | 40.2 | 60.7 | 57.3 | 69.4 | 40.0 | 60.5 | 57.1 | 58.1 | 102.8 | 29.5 |
| EcoTTA | 66.0 | 33.5 | 59.2 | 54.5 | 65.8 | 33.2 | 59.4 | 54.5 | 65.6 | 33.0 | 59.4 | 54.4 | 53.2 | 95.4 | 25.1 |
| BECoTTA | **69.5** | **45.2** | **63.5** | **59.2** | **69.4** | **44.9** | **63.5** | **59.1** | **69.4** | **44.8** | **63.5** | **59.1** | **59.3** | 98.2 | 26.8 |
| **Ours** | **69.5** | 44.1 | 62.9 | 58.7 | **69.4** | 43.9 | 62.8 | 58.5 | **69.4** | 44.0 | 62.9 | 58.6 | 58.7 | **124.5** | **38.2** |

Table 10 illustrates the trade-off between resolution and performance. Compared to the full-resolution results in the main paper, reducing the resolution to $960 \times 540$ leads to a moderate decrease in mIoU for all methods due to the loss of fine-grained details. However, our method effectively maintains robustness, achieving a mean mIoU of 58.7, which remains comparable to the state-of-the-art BECoTTA (59.3). More importantly, this setting highlights the efficiency advantage of our approach: it reaches **124.5 FPS** on the GPU and **38.2 FPS** on the Jetson Orin Nano. This substantial speedup confirms that our method is exceptionally well-suited for real-time applications on edge devices where computational budget is a critical constraint.

## A.11 Mode Switching Mechanism Ablation and Comparison

We conduct an ablation study comparing our proposed mode-switching mechanism based on environmental uncertainty ($V_{unc}$) with two widely used sample-level uncertainty metrics: *Predictive Entropy* and the *Energy Function*. This experiment evaluates the inference efficiency and overall adaptation performance under each switching strategy. The evaluation is performed on **Task 3** (Motion $\rightarrow$ Fog $\rightarrow$ Snow), and the results are summarized in Table 11.

Table 11: Ablation study of mode-switching mechanisms on Task 3.

| Switching Metric | Target Objective | Inference Speed (s/sample) ↓ | Data Upload Ratio ↓ | Mean mAP@50 (%) ↑ |
|---|---|---|---|---|
| Predictive Entropy | Sample-level Confidence | 0.45 | 65% | 39.5 |
| Energy Function | Sample-level Confidence | 0.42 | 63% | 40.1 |
| **Ours ($V_{unc}$)** | **Environment-level Drift** | **0.36** | **62%** | **40.6** |

The results clearly show that our uncertainty-based switching metric $V_{unc}$ achieves the best trade-off between efficiency and accuracy. Specifically, it attains the fastest inference speed (0.36 s/sample), the lowest data upload ratio (62%), and the highest Mean mAP@50 (40.6%). This demonstrates that $V_{unc}$ effectively captures environment-level drift, thereby avoiding unnecessary or noisy mode switching triggered by single corrupted samples—a limitation inherent to sample-level metrics such as entropy and energy.

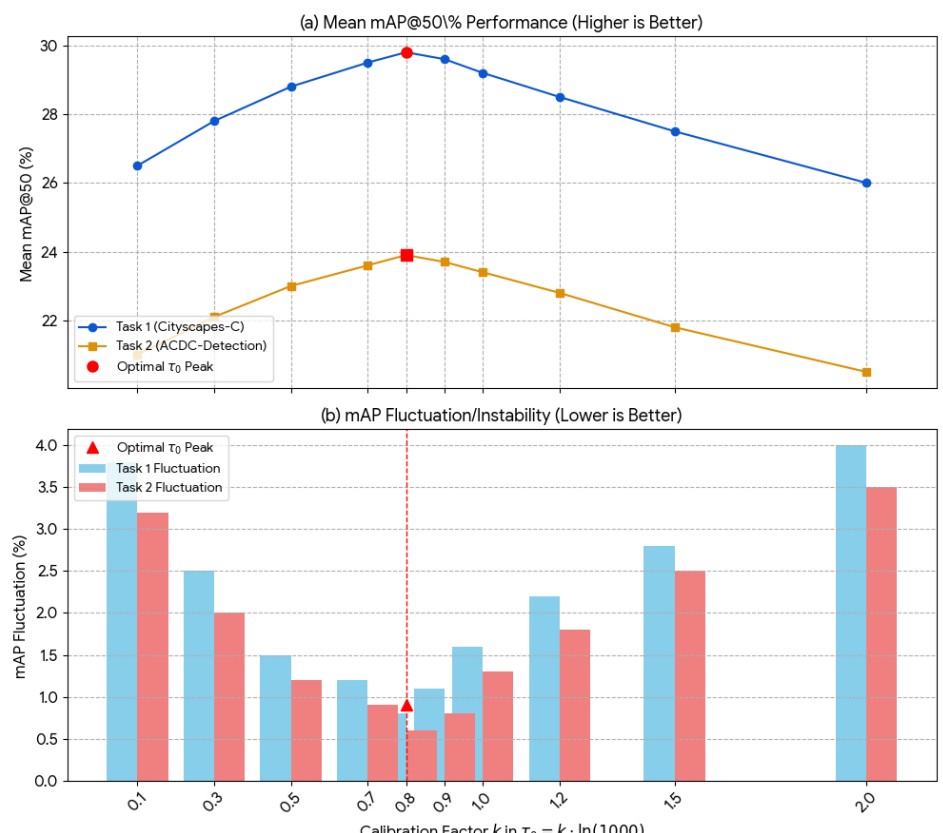

Figure 10: Ablation Study of $\tau_0$ Calibration in QuTTA. (a) Mean mAP@50% performance and (b) mAP Fluctuation (instability) are shown as a function of the calibration factor $k$.

## A.12 ABLATION STUDY OF $\tau_0$

We conducted an in-depth ablation study on the core calibration parameter $\tau_0$ within the local self-adaptation method QuTTA to verify its effectiveness in controlling pseudo-label quality and ensuring adaptation stability. We adopted the flexible, loss-based qutta $\omega(\mathbf{x})$ to replace traditional hard pseudo-label filtering, where $\tau_0$ controls the loss transition point the function. We evaluated the comprehensive performance and stability of QuTTA on Task 1 and Task 2 by varying the calibration factor $k$ ($\tau_0 = k \cdot \ln(1000)$).

Figure 10 (a) clearly shows a pronounced impact of the $\tau_0$ value on the average performance. Our setting of $\tau_0 = 0.8 \ln(1000)$ achieved the highest mAP across both tasks (Task 1: $29.8\%$; Task 2: $23.9\%$). Performance significantly degraded when $\tau_0$ deviated from the optimum. Strict settings erroneously suppressed many useful moderate-quality samples, while loose settings introduced too many high-loss noisy samples, both impacting training efficiency and accuracy. Furthermore, Figure 10 (b) illustrates the impact of $\tau_0$ on model **stability**. At the performance peak of $k = 0.8$, the mAP fluctuation simultaneously reached its **minimum** (Task 1: $0.8\%$; Task 2: $0.6\%$). Both overly strict (fluctuation up to $4.0\%$ at $k = 0.1$) and overly loose $\tau_0$ settings caused severe performance oscillations during continuous adaptation, confirming that inaccurate $\tau_0$ disrupts stability.

