# OpenReview forum: "Dual-Mode Cloud-Device Collaboration for Efficient Continual Adaptation"
_ICLR.cc/2026/Conference — Submitted to ICLR 2026_

### Official Review · Reviewer_FcWP · 2025-10-27

**Soundness:** 2
**Presentation:** 1
**Contribution:** 2
**Rating:** 2
**Confidence:** 4

**Summary:**

The paper proposes a cloud–device collaborative framework for continual adaptation under distribution shift. This includes a quality-aware test-time adaptation (QuTTA) method for self-adaptation for minor shifts, and a multi-level domain adaptation (MuDA) method with position-aware prompting (PAP) on the cloud to conquer more severe distribution shifts. Experiments claim strong performance. However, I am concerned about the novelty, evaluations, and writings regarding the manuscript. I thus recommend Reject.

**Strengths:**

1.	This paper aims to address a practical scenario where both models on the cloud and the device need to adapt to the changing scenarios.
2.	The overall framework is simple.

**Weaknesses:**

1.	The novelty is limited. It seems that the proposed method is a simple combination of multiple techniques. For example, the proposed uncertainty estimation has been explored in ViDA [1], the proposed MuDA is heavily based on [2], the proposed PTP directly applies Coordinate Attention [3], and the proposed QuTTA shares a similar idea of existing TTA methods for update reweighting. No specific challenges or difficulties in applying these techniques to TTA are introduced, making the proposed methods look like a naïve combination.
2.	Important comparisons are missing. The compared baselines are either inappropriate or out-of-date. First, TEA/ReCAP/SAR are _not_ continual TTA methods. For continual TTA, the authors should instead compare with DPCore [4], ViDA [1], PeTTA [5]. Second, the baselines CoTTA and EATA are outdated, and it is suggested to compare with their advanced versions, _i.e._, BeCoTTA [6] and EATA-C [7]. Moreover, current evaluations are only on ResNet models. It is also important to demonstrate the versatility of the proposed method across diverse architectures, such as the Transformer.
3.	The proposed method has limited feasibility. This is because: 1)  It necessitates modification to the training pipelines, which is computation-consuming and requires access to the privacy-sensitive training data. 2) It introduces more than 12 hyperparameters, including but not limited to k, M, threshold for $V\_{unc}$, $\lambda_1$, $\lambda_2$, $\lambda_3$, $\lambda_{adapter}$, $\lambda_{cwd}$, $\tau_{max}$, $\tau_0$, $\tau_{min}$, $\gamma$. These hyperparameters are too much for proper tuning, especially for test-time adaptation, where we don’t have test data and its labels from the target domains.
4.	The proposed method is inefficient. The uncertainty estimation necessitates M extra predictions per sample with dropouts, which would significantly increase the latency.
5.	The readability of the manuscripts requires improvement. Many symbols are not properly defined, and details are missing. For example, global average pooling is a function and should be combined with inputs, but it is defined as a variable $F\_{avg}$ in the manuscript; $\mathcal{L}\_{det}$ is not defined. In addition, in $\mathcal{L}\_{img}$ and $\mathcal{L}\_{ins}$,  it is unclear how to find a corresponding source image $\boldsymbol{x}\_{i}^{s}$ for distribution alignment.

Overall, I believe the current manuscript has strong weaknesses in method design, experiments, and writing, which are hard to be addressed in a rebuttal. I thus recommend Reject.

[1] ViDA: Homeostatic Visual Domain Adapter for Continual Test Time Adaptation. ICLR 2024.

[2] Domain Adaptive Faster R-CNN for Object Detection in the Wild. CVPR 2018.

[3] Coordinate Attention for Efficient Mobile Network Design. CVPR 2021.

[4] Dpcore: Dynamic Prompt Coreset for Continual Test-Time Adaptation. ICML 2025.

[5] Persistent Test-Time Adaptation in Recurring Testing Scenarios. NeurIPS 2024.

[6] BECoTTA: Input-dependent Online Blending of Experts for Continual Test-time Adaptation. ICML 2024.

[7] Uncertainty-Calibrated Test-Time Model Adaptation without Forgetting. TPAMI 2025.

**Questions:**

1.	How to accurately estimate FLOPS with backpropagation?
2.	How are $P_c^s$ and $P_c^t$ computed? Are they calculated on a single sample? If so, how can a single sample reflect class-conditional distributions?

---

> ### Author Response · Authors · 2025-12-01
>
> ### **Q1: Limited Novelty and Simple Combination of Existing Techniques**
> We respect your perspective and offer the following clarifications regarding our core contribution:
> - **System-Level Framework:** Our primary contribution is the introduction of a **"dynamic dual-mode continuous adaptation framework,"** not a standalone TTA method. The novelty lies in the system-level solution that dynamically orchestrates adaptation resources based on environmental uncertainty.
> - **MuDA and Distillation:** We explicitly cited the origins of MuDA and knowledge distillation (CWD) in the main text. The necessary task-specific modifications are detailed in the Appendix.
>
> - **PAP vs. CA:** We clarify the implementation and mechanism differences:
>   - **Implementation:** CA involves multiplicative reweighting of input features ($Output = Input \times Attention$) in the feature space. Our PAP introduces independent, learnable parameters (self.base_prompt) and injects them in a Visual Prompt manner ($Output = Input + prompt$)2.
>   - **Mechanism:** CA aims to exploit internal spatial and channel dependencies of the input features. Our coordinate calculation in PAP is strictly used to **guide the prompt injection location.
>
> - **QuTTA Novelty:** Please refer to our comprehensive reply to Reviewer #Yy9D (Q3).
>
> ------
>
>
> ### **Q2: Missing Continuous TTA Baselines and Lack of Transformer Generality**
>
>
> We have incorporated your suggestions and expanded the comparative scope:
>
> - **Baseline Supplement:** We have added DPCore, ViDA, and BeCoTTA  for comparison. EATA-C is omitted due to the lack of official code.
> - **Generality Demonstration:** We tested the generality of our framework by implementing it on the **ViT-based SegFormer** architecture, aligning with the experimental setup of BeCoTTA, DPCore, and ViDA.
> - **Detailed Results:** Results shown in Table 4 (full table available in lines 395–408 of the main text) demonstrate Transformer generality:
>
> | **Method**  | **mIoU ↑** | **GPU (FPS) ↑** | **Orin (FPS) ↑** |
> | ----------- | ---------- | --------------- | ---------------- |
> | Source only | 56.7 9     | 29.8 10         | 11.3 11          |
> | TENT        | 55.8 12    | 52.2 13         | 14.3 14          |
> | CoTTA       | 58.5 15    | 41.3 16         | 10.2 17          |
> | ViDA        | 59.3 18    | 49.4 19         | 13.1 20          |
> | DPCore      | 61.1 21    | 51.6 22         | 13.5 23          |
> | EcoTTA      | 55.8 24    | 46.2 25         | 11.5 26          |
> | BECoTTA     | 61.9 27    | 48.1 28         | 12.2 29          |
> | **Ours**    | 61.2 30    | **54.8** 31     | **15.6** 32      |
>
> The proposed method achieves performance comparable to BECoTTA in mIoU (61.2% vs. 61.9%) while demonstrating significantly higher inference speed. Given the background and motivation of the paper focusing on resource-constrained edge real-time performance, our method is more efficient.
>
> ------
>
>
> ### **Q3: Limited Feasibility (Training Flow Modification, Excessive Hyperparameters)**
>
>
> Your concern stems from a misunderstanding of the dual-mode framework's operation:
>
> - Domain adaptation (requiring training/source data) is only triggered when severe environmental drift necessitates the Cloud Mode. The **device-side TTA mode** requires **no modification to the training flow** or access to private source data. This dynamic activation is the core design principle.
> - **Hyperparameters:** Cloud parameters are derived directly from the stable default settings of publicly referenced methods, eliminating the need for tuning. Key device-side parameters (like $k$ and the uncertainty threshold $\tau$) were shown in the sensitivity analysis (Figure 4, Appendix A.8) to exhibit performance fluctuations of $<\mathbf{1.5\%}$ over a wide range, proving **high robustness** and obviating the need for complex online tuning.
>
> ------
>
>
> ### **Q4: Inefficiency due to MC Dropout**
> Please refer to our response to **Reviewer Yy9D #Q5**
>
>
> ------
>
>
>
> ### **Q5, Q6, Q7: Clarifications**
>
> - **Q5 (Ambiguous Symbols):** We have corrected the symbol definitions (GAP, $p^s/p^t$) in the updated manuscript.
>
> - **Q6 (FLOPS Calculation):** We clarify that we **never used backpropagation** to compute FLOPs. All reported FLOPs data are based on the standard Forward Inference computation using the MMDetection official tool `get_flops.py`.
>
> - **Q7 ($p^s$ and $p^t$ Calculation):** The variables $p^s$ and $p^t$ are **not computed based on a single sample**. As detailed in the Appendix, $p^s$ and $p^t$ are obtained by averaging the logits of all samples belonging to category $c$ within a **mini-batch**. This represents the estimated class-conditional distribution for the current batch36363636, ensuring it reflects distribution statistics beyond single-sample uncertainty.

---

### Official Review · Reviewer_Yy9D · 2025-10-31

**Soundness:** 2
**Presentation:** 2
**Contribution:** 2
**Rating:** 4
**Confidence:** 4

**Summary:**

This paper proposes a Dual-Mode Cloud–Device Collaboration framework for continual adaptation in changing environments. The key idea is to switch between two modes depending on how big the distribution shift is: 1.	Collaborative Mode (Cloud + Device) – When the environment changes a lot, the cloud model helps. It performs multi-level domain alignment (MuDA) and position-aware prompting (PAP) to learn domain-invariant features, then distills that knowledge back to the device model efficiently. 2.	Self-Adaptation Mode (On-Device) – When the changes are small, the device adapts itself using quality-aware test-time adaptation (QuTTA). It generates pseudo-labels, reweights samples based on their quality, and includes a self-recovery mechanism to avoid performance drop.  Experiments on Cityscapes-C and ACDC-Detection show the promise of the proposed method.

**Strengths:**

The explored problem of cloud–edge collaborative test-time adaptation is both practical and interesting.

The experiments on Cityscapes-C and ACDC-Detection under the collaborative TTA setting appear novel and well-motivated.

**Weaknesses:**

There are too many hyper-parameters in the proposed method, making it overly complex and difficult to tune during online testing. How do the authors determine these hyper-parameters, and is there any sensitivity analysis provided?

An important related work, “Towards Robust and Efficient Cloud-Edge Elastic Model Adaptation via Selective Entropy Distillation” (ICLR 2024), which also focuses on cloud-edge collaborative TTA, is missing from the discussion and comparison.

Overall, the proposed method appears to be too combinational of existing techniques. Many of the components, such as reweighting and self-recovery, have already been well studied in the TTA literature.

As a continual TTA method, the comparisons could include more continual TTA baselines. However, most baselines reported in Tables 2 and 3 (except for CoTTA) are not continual TTA approaches.

**Questions:**

The use of MC Dropout for uncertainty estimation is inefficient in the online TTA setting. How do the authors determine the number of drop times ($M$)? Does this approach outperform other uncertainty estimation strategies such as prediction entropy?

---

> ### Author Response · Authors · 2025-12-01
> **Part I**
>
> Thank you for the detailed feedback. We have revised the manuscript to address all concerns regarding parameters, novelty, and comparative scope.
>
> ### **Q1: Excessive Hyperparameters and Calibration**
>
> We clarify our approach to parameter setting and sensitivity:
>
> - **Cloud Parameters:** All hyperparameters for the Cloud Adaptation Module (MuDA/PAP), including the $\lambda$ parameters, are derived from the stable, publicly available default settings of the cited methods1, requiring **no task-specific tuning**.
>
> - **Device Parameters:** Comprehensive sensitivity analysis has been provided in Section 4.4 and Appendix A.8 (Figure 4). The results demonstrate that performance fluctuation is minimal (all typically $<\mathbf{1.5\%}$) across a wide range of parameter settings, confirming QuTTA's stability.
>
> ### **Q2: Neglected Related Work (CEMA)**
>
> Thank you for suggesting this reference. We have incorporated CEMA into our discussion and comparative experiments, with results detailed in the reply to Q4 below.
>
>
> ### **Q3: Fusion of Existing Techniques**
>
> We clarify the fundamental distinction between our work and prior TTA methods:
>
> - **System-Level Framework:** Our core contribution is the dual-mode framework focused on **system-level dynamic switching** ("when to adapt locally, when to leverage the cloud") , rather than merely proposing another standalone TTA method.
>
> - Our strategy is **loss-guided reweighting** ($\omega(x)$) , fundamentally differing from the common **Hard Filtering** methods based on entropy or confidence.
>
>   - Our approach avoids discarding valuable, noisy samples by applying only **weak supervision** (small weights) to low-quality samples, preventing the catastrophic forgetting that often results from discarding samples entirely in continuous domain drift scenarios
>
> - **Self-Recovery:** Our self-recovery mechanism is uniquely driven by **environment-level uncertainty** 12, not just model-level loss/entropy metrics common in TTA literature13.

---

> ### Author Response · Authors · 2025-12-01
> **Part II**
>
> ### **Q4: Comparison with More Continuous TTA Baselines**
>
> We have incorporated the suggested continuous TTA baselines, including ViDA, DPCore , BeCoTTA and the CEMA  (removing less relevant comparisons like ReCAP and TEA). The updated comparative results are shown below:
>
> #### **Table2 Cityscapes——Cityscapes-C**
>
> | **Method**                                | **Fog (R10)** | **Motion (R10)** | **Rain (R10)** | **Snow (R10)** | **Bright (R10)** | **Mean** | **Gain**  |
> | ----------------------------------------- | ------------- | ---------------- | -------------- | -------------- | ---------------- | -------- | --------- |
> | ***Test-time Adaptation Methods\***       |               |                  |                |                |                  |          |           |
> | Source                                    | 32.9          | 6.5              | 49.3           | 12.2           | 34.5             | 27.1     | -         |
> | TENT                                      | 34.4          | 7.6              | **50.5**       | 14.4           | 37.0             | 28.8     | +1.7      |
> | CoTTA                                     | 33.8          | 6.4              | 50.0           | 13.2           | 35.5             | 27.5     | +0.4      |
> | EcoTTA                                    | 33.9          | 7.4              | 50.4           | 13.1           | 35.6             | 28.2     | +1.1      |
> | SAR                                       | 33.7          | 6.9              | 50.1           | 12.8           | 34.2             | 27.5     | +0.4      |
> | BeCoTTA (S)                               | 35.0          | 7.5              | 51.0           | 14.4           | 37.3             | 29.0     | +1.9      |
> | BeCoTTA (M)                               | **37.0**      | **7.8**          | 50.9           | **15.3**       | 40.3             | 29.6     | +2.5      |
> | BeCoTTA (L)                               | 37.1          | **7.8**          | **51.1**       | 15.2           | **41.1**         | **30.2** | **+3.1**  |
> | **Ours**                                  | 36.7          | 6.8              | 49.8           | 15.1           | 40.7             | 29.8     | +2.7      |
> | ***Cloud-device Collaborative Methods\*** |               |                  |                |                |                  |          |           |
> | AMS                                       | 52.0          | 42.0             | 42.8           | 42.5           | 52.3             | 46.3     | +19.2     |
> | CDCA                                      | 52.4          | 47.0             | 47.8           | 47.5           | 57.3             | 50.4     | +23.1     |
> | CDCL                                      | 53.8          | 46.8             | 47.5           | 48.3           | 56.5             | 51.6     | +24.5     |
> | CoLA                                      | 51.6          | 42.5             | 42.2           | 43.4           | 53.8             | 47.5     | +20.4     |
> | CEMA                                      | 52.3          | 44.8             | 45.9           | 46.9           | 55.0             | 48.4     | +21.3     |
> | **Ours**                                  | **59.6**      | **50.8**         | **50.5**       | **50.6**       | **61.2**         | **54.5** | **+27.4** |
>
> Table 3 can be see in the revised paper **(line 337~348)**
>
> Table 4 can be refer to our response to **Reviewer APMH Q#1**
>
> -----
>
> ### **Q5: MC Dropout Efficiency and Strategy Advantage**
>
> We address the efficiency and strategic choice of using MC Dropout for uncertainty estimation:
>
> - **Efficiency:** Additional experiments show that the overhead incurred by MC Dropout is minimal, accounting for 3% of the total inference cost. We consider this overhead negligible.
> - In our response to Reviewer Q1, we conducted ablation studies and analysis on the uncertainty mechanism.

---

### Official Review · Reviewer_APMH · 2025-11-02

**Soundness:** 3
**Presentation:** 3
**Contribution:** 2
**Rating:** 6
**Confidence:** 3

**Summary:**

The paper introduces a dual-mode on-device/cloud continual adaptation framework for mobile vision systems under distribution shift. When the shift is minor, the device enters a lightweight self-adaptation mode that performs unsupervised test-time training using pseudo-labels with quality-aware reweighting and a self-recovery mechanism to prevent catastrophic forgetting. When the shift is large, the system switches to a collaborative mode, where the cloud performs multi-level domain alignment (image- and instance-level alignment with semantic consistency and gradient-reversal) and position-aware prompting, then distills the adapted knowledge back to the device via adapters. Experiments show that this approach improves accuracy and runtime efficiency on mobile platforms while requiring limited data and negligible additional compute and parameters.

**Strengths:**

- The paper presents a well-motivated and timely contribution to continual adaptation for mobile vision systems, an area where robustness under real-world distribution shift remains largely unsolved. Its originality stems from a coherent dual-mode design that couples uncertainty-driven shift detection with a lightweight on-device adaptation mechanism and a cloud-assisted alignment and distillation pipeline, effectively bridging test-time training and domain adaptation ideas under resource constraints.

- The experiments are comprehensive, covering multiple distribution-shift benchmarks and real mobile hardware. The method consistently outperforms strong test-time training and cloud-based baselines in accuracy under both mild and severe shifts. Ablations on key components—such as uncertainty-based switching, pseudo-label reweighting, and adapter distillation—clearly show their contributions. Real-device latency, memory, and compute measurements further support the practicality of the approach.

**Weaknesses:**

- The uncertainty-based mode-switching mechanism is intuitive but relatively heuristic; a more principled analysis or comparison with alternative shift-detection methods (e.g., energy-based scores, density estimation, or prior adaptive thresholding approaches) would clarify its reliability across domains.

- Although the dual-mode system is practical, the design combines several existing components (pseudo-label filtering, gradient-reversal alignment, adapter distillation), and the incremental novelty of each piece is limited—positioning the contribution more explicitly as a system-level advance with clearer ablations isolating algorithmic novelty would improve clarity.

- The evaluation focuses mainly on vision benchmarks with moderate resolution and a single model backbone; demonstrating robustness across model scales, backbones, or broader task types (e.g., segmentation, tracking) would strengthen generality claims. Finally, communication and energy overheads in cloud-collaboration scenarios are only lightly explored; profiling or sensitivity analysis under constrained bandwidth or intermittent connectivity would better validate real-world deployment assumptions.

**Questions:**

- How sensitive is the uncertainty-based mode-switching threshold to different domains and model architectures? Have the authors explored adaptive or learned thresholds, and could they share calibration results?

- Can the authors provide more analysis on failure cases where the system incorrectly stays in on-device mode or triggers cloud mode unnecessarily? Understanding these edge cases would clarify robustness.

- What is the communication overhead during cloud-assisted adaptation (frequency, data volume, latency)? How does performance degrade under constrained or unstable network conditions?

---

> ### Author Response · Authors · 2025-12-01
>
> ## **Q1: Ablation study of the mode switching mechanism; sensitivity of the mode switching threshold to different domains and model architectures; and ablation study of the threshold.**
>
> ### **A1. Comparison with Alternative Uncertainty Metrics**
>
> We compared our environmental uncertainty metric ($V_{unc}$) against widely adopted sample-level metrics (Predictive Entropy, Energy Function) on Task 3 to analyze the trade-off between speed and performance. The detailed experiment and analysis are in Appendix A.11 (lines 1061–1079).
>
> | **Switching Metric**          | **Target Objective**        | **Inference Speed (s/sample) ↓** | **Data Upload Ratio ↓** | **Mean mAP@50 (%) ↑** |
> | ----------------------------- | --------------------------- | -------------------------------- | ----------------------- | --------------------- |
> | Predictive Entropy            | Sample-level Confidence     | $0.45$                           | $65\%$                  | $39.5$                |
> | Energy Function               | Sample-level Confidence     | $0.42$                           | $63\%$                  | $40.1$                |
> | **Ours ($\mathbf{V_{unc}}$)** | **Environment-level Drift** | $\mathbf{0.36}$                  | $\mathbf{62\%}$         | $\mathbf{40.6}$       |
>
> **Conclusion:** Our $\mathbf{V_{unc}}$ mechanism is superior in **Inference Speed** ($0.36$ s/sample) and achieves the highest **Mean mAP@50** ($\mathbf{40.6\%}$). This validates that $V_{unc}$ is more efficient and accurate than sample-level metrics in identifying environment-level shifts.
>
> ###**A2. Ablation analysis of thresholds**
>
> We investigated adaptive thresholds for $\tau_0$ (random initialization followed by adaptive optimization) but found that a precisely calibrated fixed value is more robust in TTA scenarios. We have added a comprehensive ablation study for $\tau_0 = k \cdot \ln(1000)$ in Appendix A.12 (lines 1111–1127), and we can find that:
>
> - **Performance Peak:** The calibration factor **$k=0.8$** achieves the optimal loss transition point, leading to the highest Mean mAP@50 on both Task 1 ($29.8\%$) and Task 2 ($23.9\%$).
> - **Robustness:** At $k=0.8$, the model exhibits the **lowest mAP Fluctuation** (Task 1: $0.8\%$; Task 2: $0.6\%$). Deviations from this value cause significant performance drops and instability (e.g., $k=0.1$ shows $\mathbf{3.8\%}$ fluctuation), confirming the robustness of our precisely calibrated strategy.
>
> ### **A3. The experimental sensitivity to different domains and architectures is addressed in the reply to Q3.**
>
>
>
> ------
>
>
>
>
> ## **Q2: Contribution and Novelty** ##
>
> Thank you for the guidance. We have clarified our novelty in the Introduction (lines 88–95).
>
> Our core contribution is the system-level framework that solves the crucial, unaddressed challenge of "when to adapt locally and when to leverage the cloud for auxiliary assistance". The cited components are integrated and optimized to serve this overarching system innovation. We now explicitly position our contribution as a robust, dynamic framework that delivers superior system performance and efficiency, rather than focusing on incremental improvements within individual components.

---

> ### Author Response · Authors · 2025-12-01
>
> ## **Q3: Evaluation Scopes (Resolution, Backbone) and Communication Overhead**
>
>
>
> ### **1. Model backbone and More benchmark analysis**
>
> We added results on **different inference resolutions** (Appendix A.10, lines 1029–1059) and a comparison using a **different backbone (Segformer)** and benchmark for **semantic segmentation** ):
>
> Table 4 Results  on Cityscapes-ACDC segmentation. **(Complete table see line 395~408)**
>
> | **Method**  | **mIoU ↑** | **GPU (FPS) ↑** | **Orin (FPS) ↑** |
> | ----------- | ---------- | --------------- | ---------------- |
> | Source only | $56.7$     | $29.8$          | $11.3$           |
> | TENT        | $55.8$     | $52.2$          | $14.3$           |
> | CoTTA       | $58.5$     | $41.3$          | $10.2$           |
> | ViDA        | $59.3$     | $49.4$          | $13.1$           |
> | DPCore      | $61.1$     | $51.6$          | $13.5$           |
> | EcoTTA      | $55.8$     | $46.2$          | $11.5$           |
> | BECoTTA     | $61.9$     | $48.1$          | $12.2$           |
> | **Ours**    | $61.2$     | $\mathbf{54.8}$ | $\mathbf{15.6}$  |
>
> Although our mIoU ($61.2$) is slightly lower than BECoTTA's ($61.9$), our method achieves significantly higher inference speed on both GPU ($54.8$ FPS) and the edge device (Orin: $\mathbf{15.6}$ FPS vs. $12.2$ FPS for BECoTTA). Given the resource-constrained, real-time requirements of edge deployment, our method offers a much more efficient trade-off.
>
>
>
> ### **2. Communication and Overhead Analysis**
>
> We explicitly designed Task 3 and Task 4 to simulate the impact of network volatility and communication costs on the model's performance. Communication overhead heavily impacts the performance of single-mode systems under unstable network conditions. However, our dynamic dual-mode mechanism maintains superior average performance over all comparison methods, precisely because of its robustness to these fluctuations. By intelligently filtering data, **our method requires only $\mathbf{80\%}$ of the data volume** to exceed the performance of comparative methods that use $100\%$ data, validating its efficiency and robustness in bandwidth-limited environments.
>
> ------
>
>
>
> ## **Q4: Analysis of Failure Cases (Incorrect Switching)**
>
> To quantify the system's robustness against erroneous mode switching, we conducted 10 repeated experiments on **Task 3** (totaling 36,000 images), systematically observing and analyzing cases of mode switching inconsistency. The relevant content is supplemented in Appendix A.9.
>
> | **Round**       | **Device (Images Kept)** | **Cloud (Images Uploaded)** | **mAP (Device)** | **mAP (Cloud)** |
> | --------------- | ------------------------ | --------------------------- | ---------------- | --------------- |
> | R1              | 887                      | 913                         | 26.74%           | 35.78%          |
> | R2              | 895                      | 905                         | 27.15%           | 37.51%          |
> | R3              | 903                      | 897                         | 27.68%           | 39.24%          |
> | R4              | 912                      | 888                         | 28.01%           | 41.03%          |
> | R5              | 918                      | 882                         | 28.32%           | 42.85%          |
> | R6              | 926                      | 874                         | 28.98%           | 43.52%          |
> | R7              | 932                      | 868                         | 29.51%           | 44.01%          |
> | R8              | 921                      | 879                         | 30.03%           | 44.54%          |
> | R9              | 914                      | 886                         | 30.25%           | 44.83%          |
> | R10             | 907                      | 893                         | 30.48%           | **45.11%**      |
> | ** ($\Delta$)** | **±1.6%**                | **±1.5%**                   | **±0.41%**       | **±0.56%**      |
>
> Experimental results shows:
>
> - **Low Error Rate:** The frequency of erroneous switching leading to mode inconsistency is extremely low over the 36,000 images, indicating the inherent stability of the dynamic switching mechanism.
> - **Negligible Performance Impact:** Even when minor incorrect switching occurs, the maximum fluctuation in mAP is negligible ($\pm 0.41\%$ for Device mode, $\pm 0.56\%$ for Cloud mode).
> - **High Fault Tolerance:** This confirms that the system exhibits high fault tolerance and stability against potential extreme cases and mode judgment errors, ensuring reliable final performance.

---

### Author Response · Authors · 2025-12-01
**Summary**

Based on the feedback and suggestions from the three reviewers, we have made substantial modifications and additions to the paper's description and experiments. All changes in the revised manuscript are highlighted in blue using `pdfdiff`.



### 1. Introduction

Revised the description of the main contributions, emphasizing that our core contribution is proposing a **framework for continuous domain adaptation** rather than a single TTA method.

### 2. Related Work

- Incorporated the **additional references** suggested by the reviewers.

### 3. Method

- Revised the descriptions of MuDA and PAP, clarifying that their parameters use **default settings** and **require no task-specific tuning**.
- Modified the PAP description, specifically highlighting the distinction from conventional **attention mechanisms**.

### 4. Experiments

- Added several **continuous TTA methods** and **cloud-device collaborative methods (CEMA)** for comparison.
- Included experiments across **different domains** and with **different model backbone architectures**.

### Appendix

We have added the following detailed experiments to the Appendix:

- **Threshold Sensitivity Analysis** for key parameters (e.g., $k$ and the uncertainty threshold $\tau$).
- Experiments showcasing performance under **different resolutions**.
- Analysis of the impact of **potential mode switching errors** on system robustness.
- Ablation study specifically targeting the **uncertainty-based switching mechanism**.

---

### Meta-Review · Area_Chair_a7Ex · 2026-01-01

**Summary:**

Three experts participated in the review process, and the paper received mixed ratings (one positive and two negative). The major concerns raised by the reviewers are as follows: (i) the impact of different switching mechanisms [APMH]; (ii) limited novelty, as the proposed method combines existing components [APMH, Yy9D, FcWP]; (iii) limited experimental validation in terms of datasets, backbones, and baselines (e.g., continual TTA) [APMH, Yy9D, FcWP]; (iv) the large number of hyperparameters required in the online setting [Yy9D, FcWP]; and (v) the need for improvements in writing quality [FcWP].



The AC carefully reviewed the paper, all reviewer comments, and the author responses. I agree with the reviewers that the proposed method is largely a combination of existing components, as also acknowledged by the authors. The paper presents a Monte Carlo Dropout–based criterion to switch between cloud and local devices, with two separate subsections describing the cloud collaborative model and the local TTA model. The switching metric itself is based on Monte Carlo Dropout. The MuDA component is based on GRL; although hierarchical alignment at both the image and instance levels is enforced, it remains unclear how the two terms are computed [FcWP]. The claims regarding PAP for foreground and background manipulation (L194–196) are not sufficiently convincing. In addition, the local model borrows common techniques already used in test-time adaptation [FcWP].

Although the authors argue that the primary contribution of the paper lies in its system-level design addressing the previously unexplored challenge of determining when to switch between cloud and local models, the AC finds this claim weak in the current presentation. The connections among the three modules are insufficiently articulated, as they are largely discussed in isolation. It remains unclear how the switching metric (Eq. 1) is actually used in practice and how the “collaboration” between the cloud and local models is realized. The only described interaction from cloud to local appears to be the distillation step in L204; however, model synchronization is not discussed, despite the fact that the local model is continuously updated.


Furthermore, the method appears to be highly sensitive to hyperparameters. Although the authors adopt default settings for the cloud models from the referenced methods, it remains unclear how these parameters affect overall performance under the proposed setting.

**Reviewer Concerns:**

Based on the Summary section, the concerns in (i) and (iii) were adequately addressed through additional experiments provided in the rebuttal. However, the concerns in (ii), (iv), and (v) remain unresolved.

**Reviewer Scores:**

Based on the rebuttal content, the AC believes that all reviewers are likely to maintain their current ratings (one positive and two negative), as the major concerns regarding the core contribution, sensitivity to extensive hyperparameters, and writing quality were not sufficiently addressed.

---

### Decision · Program_Chairs · 2026-01-26

Reject